# MPFM: Cross Multi-Domain Prototype Flow Matching for Log Anomaly Detection

**Jing Zhang** [1]    **Chao Luo** [1]

## Abstract

Cross multi-domain log anomaly detection aims to train a unified model applying in multiple heterogeneous systems, alleviating the annotation cost and scalability bottlenecks of traditional cross single-domain approaches. However, existing methods face two fundamental challenges: (i) geometric proximity alone is insufficient to certify normality, and (ii) forcibly aligning distributions across domains can induce negative transfer. To address these issues, we propose MPFM (Cross Multi-Domain Prototype Flow Matching for Log Anomaly Detection), grounded in the principle that anomalies are samples that cannot be stably generated by the normal data-generating mechanism. Specifically, MPFM employs a shared–private prototype system to disentangle cross-domain commonalities from domain-specific patterns, introduces domain-conditioned flow matching to perform anomaly detection by integrating structural and dynamical evidence, and further leverages prototype-drift-driven hard example mining to improve robustness near decision boundaries. Experiments on HDFS, BGL, Thunderbird, and Spirit demonstrate that MPFM delivers substantial gains under multi-domain joint training.

## 1. Introduction

Modern large-scale software systems generate massive volumes of system logs during operation (Zhu et al., 2023; He et al., 2023). As a critical medium for recording system states and event sequences, log data plays a vital role in failure diagnosis and reliability assurance (Le & Zhang, 2021; Luo et al., 2026). However, with the rapid expansion

of system scale and architectural heterogeneity, log data increasingly exhibits strong multi-domain characteristics: logs generated by different systems vary significantly in syntactic formats, semantic patterns, and anomaly distributions (Zhou et al., 2025; Mylläri et al., 2025).

Most existing log anomaly detection methods, such as DeepLog (Du et al., 2017) and LogBERT (Guo et al., 2021), adopt a one-model-per-system paradigm, where data collection, annotation, and model training are conducted independently for each target system (He et al., 2025). This paradigm faces fundamental scalability issues as the number of systems grows: annotation costs increase linearly, model maintenance becomes burdensome, and knowledge learned from one system cannot be effectively reused by others (Chen et al., 2020; Han & Yuan, 2021).

These limitations have motivated the exploration of alternative modeling paradigms that leverage logs from multiple systems to train a unified model, enabling many-to-many cross-domain knowledge transfer and reuse (Ye et al., 2025; Khanal et al., 2025). Cross multi-domain log anomaly detection emerges as a promising direction in this context (Khanal et al., 2025; Ng et al., 2025). Compared to single-domain approaches, cross-domain joint learning allows models to capture common failure patterns shared across heterogeneous systems (Chang & Yu, 2025; Zhou et al., 2025), reduces dependence on labeled data from individual domains, and improves generalization to unseen systems (Mylläri et al., 2025; He et al., 2025). Despite its potential, achieving effective cross multi-domain generalization remains challenging due to two fundamental issues. First, the limitation of geometric distance. Methods such as LogDLR and LogMTC primarily rely on geometric criteria in feature space—e.g., reconstruction error or Euclidean distance—to identify anomalies (Zhou et al., 2025; He et al., 2025). However, in the presence of mixed multi-domain manifolds, the distribution of normal samples is often non-convex and multi-modal. Geometric proximity alone does not ensure normality: as illustrated in Figure 1, a boundary sample point, Case C located between two normal clusters may exhibit a small distance to both, yet fail to conform to the underlying generative process of normal data. Second, negative transfer induced by forced alignment. To extract domain-invariant representations, existing approaches of-

[1]School of Computer Science and Artificial Intelligence, Shandong Normal University, Jinan, China. Correspondence to: Chao Luo <luochao@sdnu.edu.cn>.

*Proceedings of the 43rd International Conference on Machine Learning*, Seoul, South Korea. PMLR 306, 2026. Copyright 2026 by the author(s).

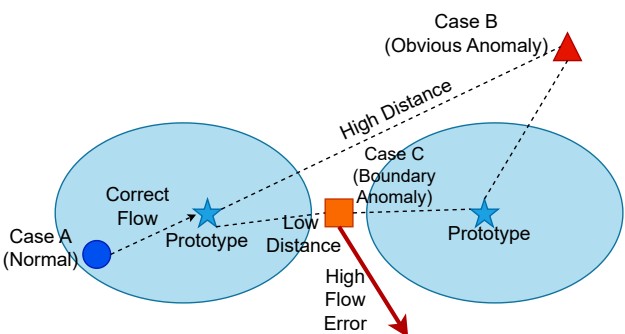

*Figure 1.* **Motivation of MPFM.** Geometric distance alone fails to detect boundary anomalies Case C. MPFM leverages flow matching error to identify such samples despite their proximity to normal prototypes.

ten employ adversarial domain adaptation or contrastive learning to aggressively align feature distributions across domains. Such alignment may distort domain-specific manifold structures, resulting in negative transfer (Sun et al., 2025; Ye et al., 2025). How to facilitate cross-domain knowledge transfer while preserving domain-specific patterns remains an open problem (Ng et al., 2025).

To tackle these challenges, we propose a flow-matching-based framework for cross multi-domain log anomaly detection. Our central insight is that anomalies are samples that cannot be stably generated by the normal data-generating dynamics. From this perspective, determining whether a sample is normal requires two types of evidence: (i) structural evidence, indicating that the sample lies near anchors of normal patterns in feature space, and (ii) dynamical evidence, indicating that the sample can be stably produced by a vector field describing the evolution of normal data.

Inspired by the strong distribution modeling capabilities of generative flow matching (Lipman et al., 2023) in sequence modeling (He et al., 2024; Wang et al., 2025) and recommendation systems (Ye et al., 2025), we introduce MPFM. Instead of relying solely on static geometric distances, MPFM learns a continuous velocity field that transports samples from a noise distribution to the manifold of normal data, thereby reframing anomaly detection as a test of consistency with learned generative dynamics.

MPFM incorporates three key innovations. First, to address cross multi-domain heterogeneity, we design a shared–private prototype system. Shared prototypes capture common failure patterns across systems, while private prototypes preserve domain-specific characteristics, structurally disentangling cross-domain commonality from individuality. Second, we introduce a domain-conditioned flow matching mechanism. Beyond mapping samples toward prototypes, the model learns a time-dependent vector field $v_t$ , such

that normal samples are generated by flowing from Gaussian noise toward prototypes. During inference, samples that cannot follow this flow—despite being geometrically close to prototypes—are identified as anomalies. Third, to strengthen the model in its weakest regions during training, we propose a prototype-drift-driven hard example mining strategy. We observe that prototype movement directions during training reveal regions of uncertainty near decision boundaries. By synthesizing virtual samples along drift directions and selecting hard examples based on flow inconsistency, we explicitly enhance robustness at distribution boundaries.

Our main contributions are summarized as follows:

- We propose MPFM, which introduces flow matching into cross multi-domain log anomaly detection and establishes a dual-decision paradigm that integrates geometric structure with generative dynamics.

- We design a shared–private prototype architecture and a drift-driven hard example mining strategy, effectively mitigating negative transfer and improving boundary detection in complex multi-domain manifolds.

- Extensive experiments on four large-scale datasets—HDFS, BGL, Thunderbird, and Spirit—demonstrate that MPFM substantially outperforms existing methods under multi-domain joint training, highlighting its potential as a foundation model for cross-domain log anomaly detection.

## 2. Related Work

### 2.1. Cross Multi-Domain Log Anomaly Detection and Prototype Learning

While single-domain log anomaly detection methods (Du et al., 2017; Guo et al., 2021; Landauer et al., 2023) achieve strong performance, they fail to generalize across systems. Recent transfer learning approaches (Han & Yuan, 2021; Chen et al., 2020; Zhao et al., 2025; Sharma & Yelleti, 2026) employ adversarial domain adaptation or meta-learning to enable cross-system detection, but fundamentally follow a source-to-target unidirectional paradigm where a pre-trained model adapts to individual target domains sequentially. This raises two issues: (1) forced alignment erases domain-specific patterns, causing negative transfer (Wang & Deng, 2018); (2) adaptation must repeat for each new domain, lacking scalability (Duan et al., 2025).

Our work differs fundamentally: we adopt a cross multi-domain joint training paradigm where domains contribute equally without source-target hierarchy. We leverage prototypical learning (Snell et al., 2017), effective in few-shot learning and OOD detection (Lee et al., 2018; Wang et al.,

2022), but existing prototype methods learn either domain-invariant (Ganin et al., 2016; Saito et al., 2018) or domain-specific (Bousmalis et al., 2016) representations in isolation. To facilitate robust cross-domain knowledge sharing, we introduce a shared-private prototype architecture: $P_{shared}$ captures universal normal patterns while $P_{domain}^d$ preserves domain-specific characteristics (e.g., Java traces vs. kernel panics), preventing negative transfer through dual-distance anomaly scoring that combines both prototype layers.

## 2.2. Generative Dynamics for Anomaly Evidence

Generative models for anomaly detection—VAEs (An & Cho, 2015), GANs (Goodfellow et al., 2014), normalizing flows (Dinh et al., 2017; Kobyzev et al., 2021; Gudovskiy et al., 2022)—model normal data distributions. However, flows can fail on OOD data by modeling low-level correlations rather than semantics (Kirichenko et al., 2020), problematic in multi-domain scenarios. Flow Matching (Lipman et al., 2023) offers simulation-free training (Chen et al., 2018), yet existing methods (Tailanian et al., 2024) use only **geometric evidence** (final likelihood), treating models as static estimators.

We propose **training dynamics as anomaly evidence**: trajectory drift $|\mathbf{v}_\theta(x_t, t, d) - \mathbf{u}_t(x|x_1)|^2$ reveals geometric blind spots where models struggle to learn stable dynamics. This enables **drift-driven hard mining**—samples with persistent high drift indicate insufficient prototype representation, even when distance metrics appear normal. Domain-conditioned flow $\mathbf{v}_\theta(x_t, t, d)$ preserves domain-specific dynamics (Song et al., 2021; Vahdat & Kautz, 2020). This **dual-evidence paradigm** combines spatial deviations (prototype distance) with temporal instabilities (flow drift).

## 3. Method

### 3.1. Overview

We redefine anomalies as samples inconsistent with normal generative dynamics, requiring dual evidence for detection: geometric proximity and dynamical stability. In Figure 2, MPFM implements this by integrating a shared-private prototype system (structure) with domain-conditioned flow matching (dynamics). Furthermore, leveraging the insight that uncertainty aligns with representation evolution, we employ prototype drift to identify and reinforce boundary blind spots via hard example mining.

### 3.2. Problem Definition

We address the problem of cross multi-domain anomaly detection across $D$ source domains. For each domain $d \in \{1, \ldots, D\}$, we have a training set $\mathcal{X}_d^{\text{train}} = \{x_i^d\}_{i=1}^{N_d}$ (consisting primarily of normal samples) and a test set $\mathcal{X}_d^{\text{test}}$ (containing both normal and anomalous samples). Let $y \in \{0, 1\}$ denote the sample label, where $y = 0$ indicates normal and $y = 1$ indicates anomalous. Each input $x \in \mathbb{R}^{T \times F}$ is a temporal sequence with $T$ time steps and $F$ feature dimensions. Our objective is to learn a unified anomaly scoring function $s : \mathbb{R}^{T \times F} \times \{1, \ldots, D\} \to \mathbb{R}$ capable of transferring shared knowledge effectively across all domains, remaining robust to distribution shifts, and making reliable decisions at the boundaries between normal and abnormal regions.

### 3.3. Shared-Private Prototype System

The core challenge in cross multi-domain scenarios lies in the duality of normal patterns: they exhibit both cross-domain commonalities and domain-specific idiosyncrasies. To address this, we design a dual-layer prototype architecture to structurally disentangle these aspects. Specifically, we maintain a set of Shared Prototypes $\mathbf{P}_s = \{p_1^s, \ldots, p_{K_s}^s\}$ to capture universal failure patterns, and sets of Private Prototypes $\mathbf{P}_d = \{p_1^d, \ldots, p_{K_p}^d\}$ for each domain $d$ to capture domain-specific features.

Given an input sequence $x$, we first extract its representation via a multi-head self-attention Transformer encoder:

$$h = \text{Encoder}(x) \in \mathbb{R}^{d_h} \tag{1}$$

where the encoder comprises $L$ Transformer blocks with $H$ attention heads. Shared prototypes are mapped to the representation space via a learnable projection $\tilde{p}_i^s = \text{Proj}(p_i^s)$, while private prototypes are further injected with a domain embedding to enhance domain distinctiveness:

$$\tilde{p}_i^d = \text{Proj}(p_i^d) + \mathbf{e}_d \tag{2}$$

where $\mathbf{e}_d \in \mathbb{R}^{d_h}$ is a learnable domain embedding vector. For a representation $h$ from domain $d$, we compute its assignment probability to the combined prototype set $\tilde{\mathbf{P}}_d = \tilde{\mathbf{P}}_s \cup \tilde{\mathbf{P}}_d$ using temperature-scaled softmax:

$$\alpha_i = \frac{\exp(-\|h - \tilde{p}_i\|_2^2/\tau)}{\sum_j \exp(-\|h - \tilde{p}_j\|_2^2/\tau)} \tag{3}$$

where $\tau > 0$ is a temperature parameter controlling the smoothness of the distribution. Subsequently, the soft-assignment weighted reconstructed representation is given by $\hat{h} = \sum_i \alpha_i \tilde{p}_i$. The reconstruction error $\|h - \hat{h}\|_2$ provides the first tier of normality evidence—whether the sample is geometrically close to the anchors of normal patterns.

### 3.4. Domain-Conditioned Flow Matching

Geometric distance alone is insufficient to fully characterize normality. Consider an anomaly falling precisely between two normal prototypes; its reconstruction error might be

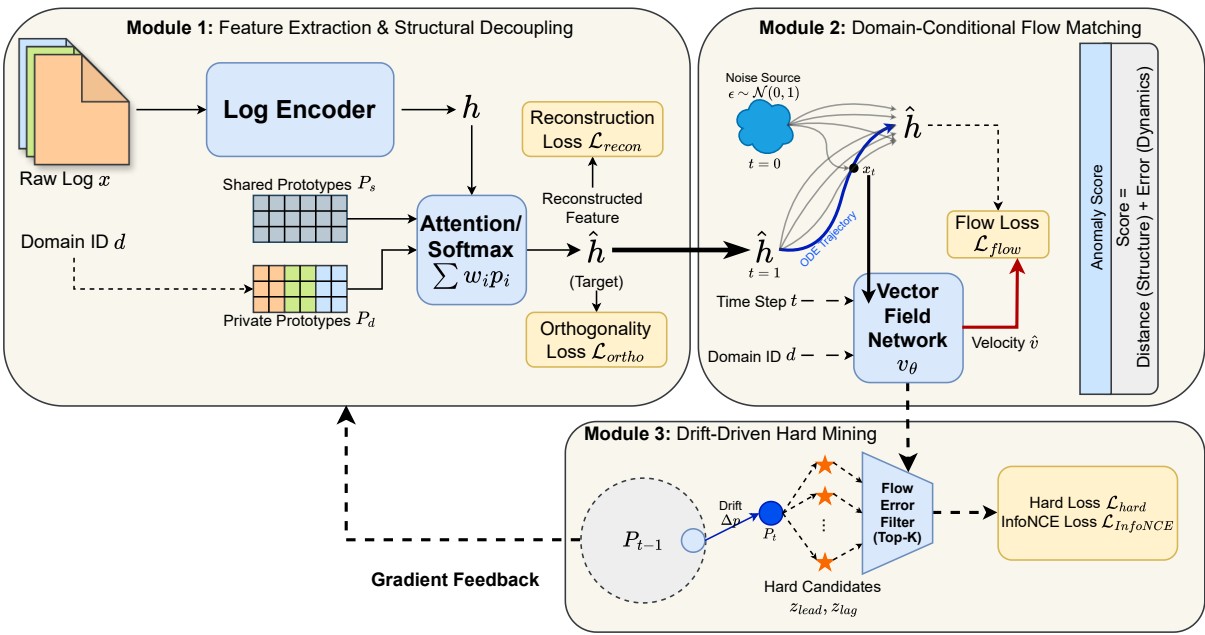

Figure 2. Three key modules: (1) Shared-private prototype system for structural decoupling; (2) Domain-conditional flow matching learning velocity field $v_\theta(t, x_t, d)$; (3) Drift-driven mining synthesizing hard examples along prototype trajectories.

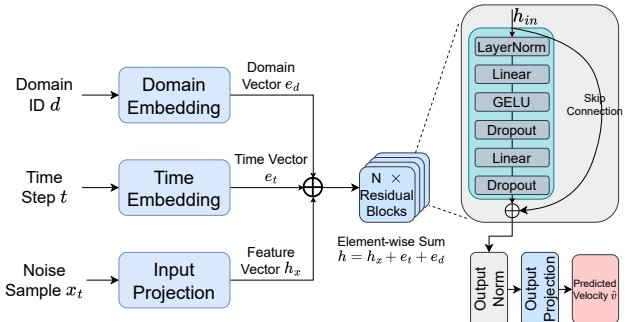

Figure 3. **Velocity Field Estimator.** Domain ID $d$ and time $t$ embeddings are fused with $x_t$, processed through $N$ residual blocks to predict $\hat{v}$.

small, yet it does not adhere to the true generative mechanism of normal data. To this end, we introduce a second tier of evidence–whether the sample can be stably produced by consistent generative dynamics.

We employ Flow Matching to model the generative process of normal data. Specifically, we define a time-dependent velocity field $v_\theta(t, x_t, d)$ that continuously transforms simple standard Gaussian noise into the target distribution:

$$\frac{dx_t}{dt} = v_\theta(t, x_t, d), \quad x_0 \sim \mathcal{N}(0, I), \quad x_1 = \hat{h} \quad (4)$$

During training, we construct the path from noise to the target via linear interpolation:

$$x_t = (1 - t)\epsilon + t \cdot \hat{h}, \quad \epsilon \sim \mathcal{N}(0, I) \quad (5)$$

where $t \in [0, 1]$ is the time variable and $\epsilon$ is a noise vector sampled from the standard Gaussian distribution. The corresponding target velocity is $v_{\text{target}} = \hat{h} - \epsilon$. The velocity field is trained by minimizing the following flow matching loss:

$$\mathcal{L}_{\text{flow}} = \mathbb{E}_{x,t,\epsilon} \left[ \|v_\theta(t, x_t, d) - v_{\text{target}}\|_2^2 \right] \quad (6)$$

This design ensures that normal samples must satisfy a dual criterion: they must be spatially close to prototypes and capable of being stably transported to their target positions by the domain-conditioned velocity field, providing a dynamical guarantee of consistency.

### 3.5. Prototype Drift-Driven Hard Example Mining

Conventional hard example mining strategies often rely on random sampling or fixed heuristics, ignoring the dynamic evolution of model weaknesses during training. We propose a two-stage mining strategy based on prototype drift and flow error.

In each training epoch, prototypes move according to gradient updates. We observe that the direction of prototype movement reveals the evolutionary trend of the representation space structure. Let $\tilde{p}^{(e)}$ be the projected prototype at the end of epoch $e$. We define the drift vector as:

$$\Delta = \tilde{p}^{(e)} - \tilde{p}^{(e-1)} \quad (7)$$

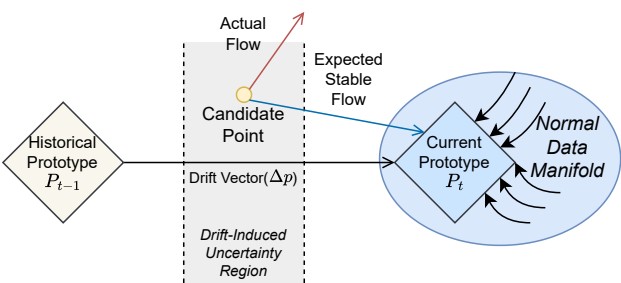

**Figure 4.** **Drift-Driven Hard Mining.** Drift vector $\Delta = P_t - P_{t-1}$ defines uncertainty region (grey). Candidates with large flow discrepancy (red vs. blue arrows) are identified as hard examples.

The region between the old and new prototypes along the drift direction represents the drift-induced uncertainty region, where the model is most prone to confusion. Figure 4 visually illustrates this mechanism. Based on this insight, we perform structured extrapolation along the drift direction to generate lag and lead candidates:

$$z_{\text{lag}} = \tilde{p}^{(e)} - \beta\Delta, \quad z_{\text{lead}} = \tilde{p}^{(e)} + \beta\Delta \quad (8)$$

where $\beta \in (0, 1)$ is a hyperparameter controlling the magnitude of extrapolation. The lag candidate approximates the historical position, while the lead candidate extrapolates along the trend. This generation method directly targets the boundary regions the model is currently learning to resolve, rather than sampling randomly.

However, not all candidates along the drift direction are equally important. We introduce flow error as a difficulty metric: if the flow field cannot accurately predict how to generate a point from noise, it implies the point lies in a current weak region of the model. For a candidate point $z$ and domain $d$, we define the difficulty score:

$$\mathcal{H}(z, d) = \mathbb{E}_{t,\epsilon} \left[ \|v_\theta(t, x_t, d) - v_{\text{target}}\|_2^2 \right] \quad (9)$$

where $x_t = (1 - t)\epsilon + tz$ and $v_{\text{target}} = z - \epsilon$. We rank all candidates for each domain by difficulty and select the top-$K$ with the highest errors:

$$\mathcal{Z}_{\text{hard}}^d = \text{Top-}K \left( \{z_i^d\}, \{\mathcal{H}(z_i^d, d)\} \right) \quad (10)$$

This two-stage strategy implements an adaptive curriculum learning process: as training progresses, prototype positions shift, drift directions update, and the model's weak regions change dynamically, with hard examples adjusting automatically.

## 3.6. Training Objectives

We jointly optimize the model using multiple complementary loss functions:

$$\mathcal{L} = \mathcal{L}_{\text{flow}} + \mathcal{L}_{\text{recon}} + \lambda_1 \mathcal{L}_{\text{InfoNCE}} + \lambda_2 \mathcal{L}_{\text{ortho}} + \lambda_3 \mathcal{L}_{\text{hard}} \quad (11)$$

where $\lambda_1, \lambda_2, \lambda_3$ are weighting coefficients. The flow matching loss $\mathcal{L}_{\text{flow}}$ trains the velocity field to learn the generative dynamics of normal data. The reconstruction loss ensures consistency between representations and prototypes:

$$\mathcal{L}_{\text{recon}} = \mathbb{E}_x \left[ \|h - \hat{h}\|_2^2 \right] \quad (12)$$

The InfoNCE contrastive loss contrasts the current representation against historical prototypes and hard candidates. For a representation $h$ and its nearest current prototype $p^+$, the negative sample set $\mathcal{N}$ includes historical shared prototypes, historical private prototypes, and the hard candidates $\mathcal{Z}_{\text{hard}}^d$. Let $s(u, v) = \text{sim}(u, v)/\tau$ denote the temperature-scaled similarity. The loss is defined as:

$$\mathcal{L}_{\text{InfoNCE}} = -\log \frac{\exp(s(h, p^+))}{\exp(s(h, p^+)) + \sum_{p^- \in \mathcal{N}} \exp(s(h, p^-))} \quad (13)$$

This design enforces explicit decision boundaries within drift-induced uncertainty regions. Orthogonal regularization prevents prototype collapse, ensuring coverage of diverse modes of the normal distribution.

$$\mathcal{L}_{\text{ortho}} = \|\mathbf{G}_s - \mathbf{I}\|_F^2 + \|\mathbf{G}_p - \mathbf{I}\|_F^2 \quad (14)$$

where $\mathbf{G}_s$ and $\mathbf{G}_p$ are the Gram matrices of the normalized shared and private prototypes, respectively. Finally, the hard flow loss imposes additional flow supervision on the mined hard samples, directly repairing weak regions:

$$\mathcal{L}_{\text{hard}} = \mathbb{E}_{z \in \mathcal{Z}_{\text{hard}}} \left[ \mathcal{H}(z, d) \right] \quad (15)$$

## 3.7. Anomaly Scoring and Calibration

During inference, we compute the anomaly score by combining dual evidence—structural distance and dynamical consistency:

$$s(x) = \min_i \|h - \tilde{p}_i\|_2 + \gamma \cdot \mathcal{H}(h, d)\big|_{t=t^*} \quad (16)$$

where $\gamma > 0$ weights the flow score, balancing the relative importance of the two evidence types. The first term measures the distance to the nearest prototype (structural evidence), while the second term measures the fitting error of the flow field at that point (dynamical evidence). The flow error is evaluated at $t^* = 0.5$, a midpoint that provides a robust default choice along the generative trajectory. We further analyze the sensitivity of $t^*$ in Appendix A.

Since score distributions may vary across domains, we apply z-score normalization to ensure threshold consistency:

$$s_{\text{cal}}(x) = \frac{s(x) - \mu_d}{\sigma_d} \quad (17)$$

where $\mu_d$ and $\sigma_d$ are estimated from normal samples in the validation set of domain $d$. The calibrated scores allow for cross-domain anomaly detection using a unified threshold.

### 3.8. Theoretical Analysis

This section establishes a theoretical framework to validate the effectiveness of MPFM. Building on the manifold distribution assumption (Cayton, 2005) and optimal transport theory (Villani, 2009), we demonstrate the limitations of relying solely on geometric distance and derive the theoretical lower bound of the flow matching error as an anomaly score.

Existing reconstruction- or distance-based methods assume normal samples cluster within compact Euclidean spheres. However, high-dimensional log data typically resides on low-dimensional manifolds (Narayanan & Mitter, 2010).

**Definition 3.1** (Normal Data Manifold). Assume the latent representation $h$ of normal data follows a distribution $P_{\text{data}}$ supported on a low-dimensional manifold $\mathcal{M}$ embedded in the ambient space $\mathbb{R}^{d_h}$.

**Lemma 3.2** (Blind Spots of Geometric Distance). *For any given distance threshold $\epsilon > 0$ and prototype set $\mathbf{P} \subset \mathcal{M}$, there exists a set of off-manifold samples $\mathcal{B}$ (i.e., blind spots) such that $\forall x \in \mathcal{B}$:*

$$\min_{p \in \mathbf{P}} \|x - p\|_2 < \epsilon \quad and \quad P_{data}(x) \approx 0 \qquad (18)$$

*Proof (Sketch).* Consider $\mathcal{M}$ as a non-convex set (e.g., a curved ribbon structure). According to manifold learning theory by Cayton (2005), Euclidean distance fails to correctly approximate geodesic distance on non-convex manifolds. Regions in the void between two adjacent branches of the manifold may have small Euclidean distances to the manifold but do not belong to it. Consequently, relying solely on geometric distance $\|h - \hat{h}\|_2$ will misclassify anomalies in $\mathcal{B}$ as normal. $\square$

MPFM addresses this by learning conditional probability paths via flow matching. According to Lipman et al. (2023), our objective is to regress a generating vector field $u_t(x)$.

**Theorem 3.3** (Flow Matching Error and Distribution Shift). *Let $v_\theta$ be the optimal velocity field minimizing the flow matching loss $\mathcal{L}_{flow}$. For any sample $x$, its flow matching error $\mathcal{H}(x)$ is related to the norm of the score function (gradient of log-likelihood). If $x$ deviates from the probability path of normal data generation (i.e., is an anomaly), a lower bound $\delta > 0$ exists:*

$$\mathbb{E}_t\left[\|v_\theta(t, \phi_t(x)) - u_t(\phi_t(x))\|_2^2\right] \geq \delta \qquad (19)$$

*Analysis.* Based on Albergo et al. (2023), the flow matching loss intrinsically measures the dynamical consistency of sample generation. For normal samples, $v_\theta$ forms a consistent vector field mapping them from noise, yielding an error approaching 0. However, for samples in the blind spot

$\mathcal{B}$ (from Lemma 3.2), although geometrically close to prototypes, their required generative trajectories conflict with or are orthogonal to the global normal vector field, resulting in significant flow error. This proves that $\mathcal{H}(x)$ is an effective statistic for detecting anomalies in geometric blind spots. $\square$

Our proposed hard example mining strategy is based on prototype drift direction. The rationale for this design is explained by gradient flow dynamics.

**Assumption 3.4** (Prototype Gradient Flow). During training, prototype updates follow the negative gradient direction of the loss function, i.e., $\Delta p^{(e)} \approx -\eta \nabla_p \mathcal{L}_{\text{total}}$.

**Theorem 3.5** (Adversarial Nature of Drift Direction). *For distance-based classifiers, the direction of prototype movement $\Delta p$ points toward regions of uncertainty in the current decision boundary. Samples generated along $\Delta p$, defined as $z = p + \beta \Delta p$, satisfy:*

$$\mathcal{L}(z) \geq \mathcal{L}(p) \qquad (20)$$

*Proof (Sketch).* Referencing the analysis of Prototype Networks by Snell et al. (2017) and adversarial training theory by Goodfellow et al. (2015), the gradient of the loss function points in the direction where model prediction confidence decreases most rapidly. Thus, the historical trajectory vector of the prototype $\Delta = p^{(e)} - p^{(e-1)}$ approximates the direction of steepest ascent on the local loss surface (which corresponds to moving away from the cluster center for normal samples). Therefore, sampling at $p^{(e)}$ along $\Delta$ is essentially approximating the search for adversarial examples on the current decision boundary. $\square$

By combining Theorem 3.3 and Theorem 3.5, MPFM's mining strategy effectively searches for samples in the tangent space of the manifold that exhibit "limited geometric distance increase" but "severe dynamical conflict," thereby efficiently reinforcing the model's robustness at distribution boundaries.

## 4. Experiments

### 4.1. Experimental Setup

**Datasets & Protocol.** We evaluate on HDFS(Xu et al., 2009; Zhu et al., 2023), BGL(Oliner & Stearley, 2007; Zhu et al., 2023), Thunderbird(Oliner & Stearley, 2007; Zhu et al., 2023), and Spirit(Oliner & Stearley, 2007; Zhu et al., 2023) under a Cross Multi-Domain Joint Training strategy, where pooled data trains a unified model evaluated on individual domains. Following LogDLR, we use Sentence-BERT embeddings (384-dim) and sliding windows ($W = 20, S = 2$) for sequence generation.

*Table 1.* Reference comparison with single-domain baselines. Baselines and MPFM-SD are trained on individual domains, while MPFM is trained jointly across all domains. Avg F1 denotes the unweighted arithmetic mean over the four datasets. Best results in each column are in bold.

| METHOD | HDFS | BGL | TB | SPIRIT | AVG F1 |
|---|---|---|---|---|---|
| DEEPLOG | 0.958 | 0.942 | 0.820 | 0.754 | 0.869 |
| LOGANOMALY | 0.962 | 0.953 | 0.855 | 0.782 | 0.888 |
| LOGBERT | 0.974 | 0.961 | 0.926 | 0.880 | 0.935 |
| LOGDLR | 0.985 | **0.982** | **0.948** | 0.945 | **0.965** |
| MPFM-SD | 0.981 | 0.968 | 0.932 | 0.921 | 0.951 |
| MPFM (OURS) | **0.987** | 0.959 | 0.915 | **0.977** | 0.960 |

*Table 2.* Performance comparison under the cross multi-domain joint training setting. All methods are trained on the pooled data from four datasets and evaluated on each domain. Avg F1 denotes the unweighted arithmetic mean over the four datasets. Best results are in bold.

| METHOD | HDFS | BGL | TB | SPIRIT | AVG F1 |
|---|---|---|---|---|---|
| LOGANOMALY | 0.756 | 0.672 | 0.791 | 0.688 | 0.727 |
| LOGTRANSFER | 0.645 | 0.443 | 0.679 | 0.581 | 0.587 |
| LOGTAD | 0.823 | 0.793 | 0.815 | 0.807 | 0.810 |
| LOGDLR | 0.852 | 0.814 | 0.779 | 0.881 | 0.832 |
| **MPFM (OURS)** | **0.987** | **0.959** | **0.915** | **0.977** | **0.960** |

**Baselines.** We compare MPFM against three categories of state-of-the-art methods: (1) RNN-based models **DeepLog** (Du et al., 2017) and **LogAnomaly** (Meng et al., 2019); (2) The Transformer-based model **LogBERT** (Guo et al., 2021); and (3) Cross-domain/Transfer learning approaches **LogTransfer** (Chen et al., 2020), **LogTAD** (Han & Yuan, 2021), and **LogDLR** (Zhou et al., 2025).

**Implementation.** Implemented in PyTorch, MPFM uses a 2-layer Transformer encoder (dim 256). We employ 32 shared and 16 private prototypes per domain. The flow matching uses an ODE solver (20 steps). Training uses AdamW ($lr = 10^{-4}$, batch 256) for 30 epochs, with drift coefficient $\beta = 0.5$ and top-32 hard mining. Additional sensitivity analyses for the main hyperparameters, including $\tau, K, \lambda_1, \lambda_2, \lambda_3, \beta$, prototype numbers, and $t^*$, are provided in Appendix A.

### 4.2. Main Results

**Comparison with Single-Domain Baselines.** Table 1 presents the performance comparison. Note that baseline methods (DeepLog, LogAnomaly, LogBERT, LogDLR) follow the standard single-domain paradigm (trained and tested on each dataset independently), whereas MPFM employs multi-domain joint training (one unified model for all datasets). This comparison verifies whether MPFM can match or exceed specialized single-domain models via joint learning.

**Analysis.** As shown in Table 1, MPFM-SD outperforms LogBERT on average and remains competitive with LogDLR under the single-domain training setting, indicating that the proposed prototype-flow architecture itself contributes to log anomaly detection rather than merely benefiting from pooled training data. Compared with MPFM-SD, the jointly trained MPFM improves performance on HDFS and Spirit, especially on the more heterogeneous Spirit dataset. However, MPFM does not uniformly outperform all specialized single-domain baselines: LogDLR still achieves higher scores on BGL, TB, and the average F1

in this reference comparison. This result suggests that the main advantage of MPFM should be interpreted in the cross multi-domain joint training setting, where a single model is required to serve multiple heterogeneous systems.

On Spirit, MPFM achieves the most pronounced improvement over the single-domain baselines. This indicates that preserving domain-specific structures while introducing dynamical consistency is particularly useful for large and heterogeneous log datasets. In contrast, the relatively lower scores on BGL and TB suggest that joint multi-domain training can still introduce interference for some domains, which further motivates the need for explicit domain-specific preservation and careful evaluation under the same joint-training protocol.

### 4.3. Cross Multi-Domain Joint Training Comparison

To fairly evaluate adaptability in cross multi-domain scenarios, we re-implement all baselines under the same joint training setting as MPFM, where training data from all four datasets are pooled to train a single model and the model is evaluated on each domain separately. For baselines originally designed with domain-adversarial components, we extend the source–target domain discriminator to a four-class domain discriminator while keeping the remaining architecture and objectives unchanged. Methods without explicit domain-adaptation modules are directly trained on the pooled multi-domain data.

As shown in Table 2, existing baselines suffer clear performance degradation when extended to the cross multi-domain joint training setting. LogAnomaly directly learns from the pooled multi-domain data and is affected by pattern confusion, where normal patterns from one domain may resemble abnormal patterns in another. LogTransfer and LogTAD are originally designed for source–target transfer, and their direct extension to four-domain joint training still lacks an explicit mechanism to preserve domain-specific normal structures. Although LogDLR mitigates domain shift through adversarial alignment, its domain-invariant representation may suppress domain-specific yet anomaly-relevant patterns

under highly heterogeneous multi-domain data. In contrast, MPFM achieves the most stable performance across all domains. This improvement is attributed to three coupled designs: (1) shared–private prototypes for structural decoupling and safe knowledge sharing; (2) domain-conditioned flow matching for learning domain-aware generative dynamics; and (3) drift-driven hard mining for reinforcing boundary robustness. These results suggest that preserving domain-specific normal structures is essential for reliable single-model multi-domain deployment. We further provide multi-seed stability results and contextual comparisons with language-model-based, zero-shot, and meta-learning baselines in Appendix B and Appendix C.

### 4.4. Efficiency Analysis

*Table 3.* Inference efficiency comparison. Batch latency is measured in milliseconds with batch size 256, and throughput is measured in sequences per second. Avg F1 is computed over the four datasets..

| Method | Steps | Batch Lat. (ms) | Thput. | Avg F1 |
|---|---|---|---|---|
| LogBERT | – | 21.0 | 12,200 | 0.935 |
| LogDLR | – | 30.1 | 8,500 | 0.832 |
| MPFM | 20 | 673.7 | 380 | 0.960 |
| MPFM | 5 | 176.6 | 1,450 | 0.948 |
| MPFM | 2 | 91.4 | 2,800 | 0.916 |

Since MPFM introduces an ODE-based flow evaluation during inference, we further analyze its computational cost and the accuracy–efficiency trade-off. All experiments are conducted on a single RTX 3090 with batch size 256.

As shown in Table 3, MPFM requires more batch-level inference time than LogBERT and LogDLR due to the ODE-based flow evaluation. However, the number of ODE steps provides a controllable trade-off between accuracy and efficiency. The 20-step setting achieves the best average F1 score, while reducing the solver to 5 steps improves throughput from 380 to 1,450 sequences per second with only a moderate F1 decrease from 0.960 to 0.948. This suggests that MPFM can be deployed with fewer ODE steps when lower latency is required. In practical log anomaly detection systems, logs are often processed in second-level or minute-level windows rather than strict per-sample real-time inference; therefore, the 5-step configuration provides a practical balance between detection performance and deployment efficiency. Additional model complexity results, including parameter count, inference memory, and training time, are reported in Appendix D.

### 4.5. Ablation Studies

To verify the necessity of structural–dynamical dual evidence and drift-driven mining, we conduct ablation stud-

ies by removing key components from MPFM, as shown in Table 4. We report ablations on three representative datasets due to space and computational constraints, covering both relatively regular logs (HDFS) and large-scale heterogeneous logs (BGL and Spirit). Avg F1 denotes the unweighted arithmetic mean over the three datasets. $\Delta$ denotes the absolute Avg F1 drop in percentage points compared with the full MPFM.

*Table 4.* Ablation study of key components. We report F1 scores on three representative datasets.

| VARIANT | HDFS | BGL | SPIRIT | AVG F1 | $\Delta \downarrow$ |
|---|---|---|---|---|---|
| MPFM (FULL) | 0.987 | 0.959 | 0.977 | 0.974 | – |
| W/O FLOW DYNAMICS | 0.975 | 0.948 | 0.921 | 0.948 | 2.6 PP |
| W/O HARD MINING | 0.969 | 0.937 | 0.941 | 0.949 | 2.5 PP |
| W/O PRIVATE PROTO | 0.965 | 0.939 | 0.912 | 0.939 | 3.5 PP |

The ablation results show that all three components consistently contribute to MPFM under joint multi-domain training. Removing flow dynamics decreases the average F1 from 0.974 to 0.948, with the most pronounced degradation on Spirit, confirming that geometric proximity alone is insufficient for detecting dynamically inconsistent boundary samples. Removing hard mining also leads to a clear performance drop, reducing the average F1 to 0.949, which suggests that drift-driven mining helps reinforce decision boundaries and improve robustness near difficult samples. The largest degradation occurs when private prototypes are removed, where the average F1 drops to 0.939. This result indicates that preserving domain-specific normal structures is particularly important for mitigating negative transfer in heterogeneous multi-domain joint training.

### 4.6. Boundary Sample Identification

To provide mechanistic evidence for the geometric blind-spot issue, we design a controlled boundary-sample analysis.

**Experimental Design.** We select normal representation pairs $(h_i, h_j)$ from different clusters and generate boundary samples via linear interpolation: $h_{\text{boundary}} = \alpha h_i + (1-\alpha)h_j$, with $\alpha \sim \mathcal{U}(0.3, 0.7)$. These samples are geometrically close to prototypes but do not necessarily follow the normal generative dynamics. We generate 500 such samples per domain and test identification recall.

**Results.** As shown in Table 5, LogBERT and LogDLR achieve relatively low recall scores of 26.6% and 41.5%, respectively, misclassifying more than half of the constructed boundary samples. MPFM (Geo Only) also achieves only 34.7% recall, indicating that geometric proximity alone is insufficient for identifying such samples. In contrast, MPFM

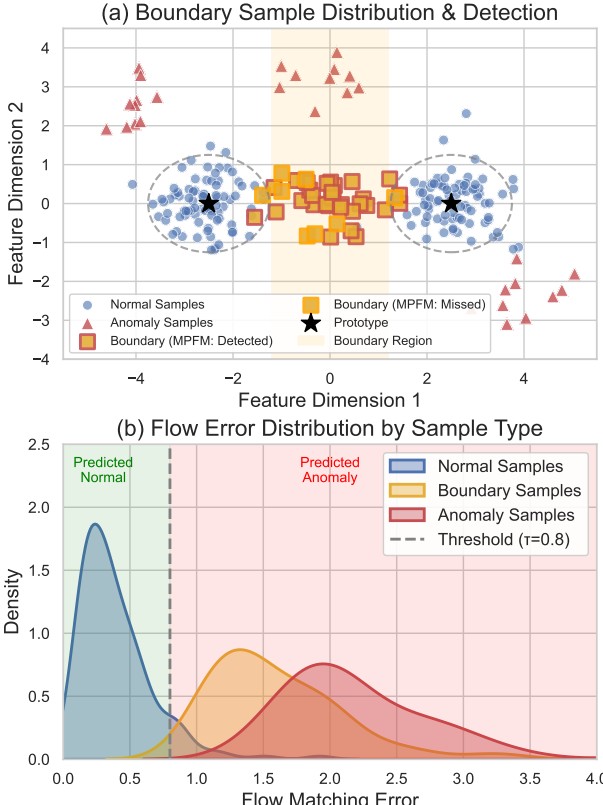

(a) Boundary Sample Distribution & Detection

(b) Flow Error Distribution by Sample Type

*Figure 5.* **Boundary Sample Identification.** (a) Feature space: normal (blue), anomaly (red), boundary (orange); red borders show MPFM detection. (b) Flow error distribution separates normal from boundary/anomaly samples.

*Table 5.* Recall rates on synthetic boundary samples. Avg denotes the average recall over HDFS, BGL, and Spirit.

| METHOD | HDFS | BGL | SPIRIT | AVG |
|---|---|---|---|---|
| LOGBERT | 0.312 | 0.287 | 0.198 | 0.266 |
| LOGDLR | 0.456 | 0.423 | 0.367 | 0.415 |
| MPFM (GEO ONLY) | 0.389 | 0.352 | 0.301 | 0.347 |
| **MPFM (FULL)** | **0.847** | **0.812** | **0.756** | **0.805** |

dence via domain-conditioned flow matching. This effectively addresses the misclassification of boundary samples common in geometric-based methods and resolves the tension between knowledge transfer and domain distinctiveness. Technically, we introduce three innovations: (1) Decoupled shared-private prototypes to handle cross-domain heterogeneity and mitigate negative transfer; (2) Domain-conditioned flow matching to provide generative dynamical evidence; and (3) Prototype-drift-driven hard example mining to adaptively reinforce decision boundaries. Experiments on four large-scale datasets demonstrate MPFM's superior performance and robust generalization in cross multi-domain settings. Future work will explore online learning mechanisms and efficient inference strategies for broader applications.

## Acknowledgements

We thank the anonymous reviewers and the area chair for their constructive feedback, and our research group members for their valuable discussions. This work was supported in part by the National Natural Science Foundation of China under Grant No. 62172264.

## Impact Statement

This work facilitates cross-domain log anomaly detection to enhance system reliability and reduce deployment costs. Potential risks include false alarms or missed detections under operational variations, and log data privacy concerns. Practical applications should therefore incorporate appropriate data governance and human oversight.

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

(Full) achieves 80.5% recall. These results support the intuition that boundary samples may remain close to normal prototypes while exhibiting high flow matching error because they are not dynamically consistent with the learned normal generative process.

Figure 5 visualizes this distinction. Boundary samples (orange squares) lie between normal clusters. While geometrically ambiguous, MPFM correctly identifies them (red borders). The error distribution in Figure 5(b) shows a clear separation: normal samples have low flow error, while boundary and anomaly samples exhibit significantly higher error, providing supporting evidence for the dual-evidence mechanism.

## 5. Conclusion

This paper proposes MPFM, a cross multi-domain prototype flow matching framework. Our core contribution is the establishment of a dual-evidence paradigm combining structural and dynamical criteria: structural evidence via a shared-private prototype system, and dynamical evi-

and Erhan, D. Domain separation networks. In *Advances in Neural Information Processing Systems*, volume 29. Curran Associates, Inc., 2016.

Cayton, L. Algorithms for manifold learning. Technical Report CS2008-0923, University of California, San Diego, 2005.

Chang, J. and Yu, F. Knowledge-guided clustering based algorithm for cross-domain recommendation problems with partially overlapping users and multiple source-domains. *Expert Systems with Applications*, 284:127936, 2025. doi: 10.1016/j.eswa.2025.127936.

Chen, R., Zhang, S., Li, D., Zhang, Y., Guo, F., Meng, W., Pei, D., Zhang, Y., Chen, X., and Liu, Y. LogTransfer: Cross-system log anomaly detection for software systems with transfer learning. In *31st IEEE International Symposium on Software Reliability Engineering, ISSRE 2020, Coimbra, Portugal, October 12-15, 2020*, pp. 37–47, 2020. doi: 10.1109/ISSRE5003.2020.00013.

Chen, R. T. Q., Rubanova, Y., Bettencourt, J., and Duvenaud, D. K. Neural ordinary differential equations. In *Advances in Neural Information Processing Systems*, volume 31. Curran Associates, Inc., 2018.

Dinh, L., Sohl-Dickstein, J., and Bengio, S. Density estimation using real NVP. In *International Conference on Learning Representations (ICLR)*, 2017.

Du, M., Li, F., Zheng, G., and Srikumar, V. DeepLog: Anomaly detection and diagnosis from system logs through deep learning. In *Proceedings of the 2017 ACM SIGSAC Conference on Computer and Communications Security*, CCS '17, pp. 1285–1298, New York, NY, USA, 2017. Association for Computing Machinery. doi: 10.1145/3133956.3134015.

Duan, C., He, M., Xiao, P., Jia, T., Zhang, X., Zhong, Z., Luo, X., Niu, Y., Zhang, L., Wu, Y., Yu, S., Hong, W., Li, Y., and Huang, G. LogAction: Consistent cross-system anomaly detection through logs via active domain adaptation, 2025. arXiv preprint arXiv:2510.03288.

Ganin, Y., Ustinova, E., Ajakan, H., Germain, P., Larochelle, H., Laviolette, F., March, M., and Lempitsky, V. Domain-adversarial training of neural networks. *Journal of Machine Learning Research*, 17(59):1–35, 2016.

Goodfellow, I. J., Pouget-Abadie, J., Mirza, M., Xu, B., Warde-Farley, D., Ozair, S., Courville, A., and Bengio, Y. Generative adversarial nets. In *Advances in Neural Information Processing Systems (NeurIPS)*, volume 27. Curran Associates, Inc., 2014.

Goodfellow, I. J., Shlens, J., and Szegedy, C. Explaining and harnessing adversarial examples. In *International Conference on Learning Representations (ICLR)*, 2015.

Gudovskiy, D., Ishizaka, S., and Kozuka, K. CFLOW-AD: Real-time unsupervised anomaly detection with localization via conditional normalizing flows. In *IEEE/CVF Winter Conference on Applications of Computer Vision (WACV)*, pp. 98–107, 2022.

Guo, H., Yuan, S., and Wu, X. LogBERT: Log anomaly detection via BERT. In *2021 International Joint Conference on Neural Networks (IJCNN)*, pp. 1–8, 2021. doi: 10.1109/IJCNN52387.2021.9534113.

Han, X. and Yuan, S. Unsupervised cross-system log anomaly detection via domain adaptation. In *Proceedings of the 30th ACM International Conference on Information & Knowledge Management*, Cikm '21, pp. 3068–3072, New York, NY, USA, 2021. Association for Computing Machinery. doi: 10.1145/3459637.3482209.

He, S., Deng, T., Chen, B., Simon Sherratt, R., and Wang, J. Unsupervised log anomaly detection method based on multi-feature. *Computers, Materials & Continua*, 76(1): 517–541, 2023. doi: 10.32604/cmc.2023.037392.

He, S., Liu, R., Chen, B., Xie, K., and Wen, J. Unsupervised multi-target cross-service log anomaly detection. *IEEE Transactions on Sustainable Computing*, 10(5): 1056–1069, 2025. doi: 10.1109/TSUSC.2025.3578517.

He, Y., Yan, T., Zhan, Y., Feng, Z., and Xia, Y. SGFM: Conditional flow matching for time series anomaly detection with state space models. *IEEE Internet of Things Journal*, 11(22):36979–36990, 2024. doi: 10.1109/JIOT.2024.34 39672.

Khanal, S., Huang, H.-Z., Huang, C.-G., Dahal, A., Huang, T., and Niazi, S. G. Domain-specific dual network with unsupervised domain adaptation for transfer fault prognosis across machines using multiple source domains. *IEEE Transactions on Instrumentation and Measurement*, 74: 1–13, 2025. doi: 10.1109/TIM.2025.3556193.

Kirichenko, P., Izmailov, P., and Wilson, A. G. Why normalizing flows fail to detect out-of-distribution data. In *Advances in Neural Information Processing Systems*, volume 33, pp. 20578–20589. Curran Associates, Inc., 2020.

Kobyzev, I., Prince, S. J., and Brubaker, M. A. Normalizing flows: An introduction and review of current methods. *IEEE Transactions on Pattern Analysis and Machine Intelligence*, 43(11):3964–3979, 2021. doi: 10.1109/TPAMI.2020.2992934.

Landauer, M., Onder, S., Skopik, F., and Wurzenberger, M. Deep learning for anomaly detection in log data: A survey. *Machine Learning with Applications*, 12:100470, 2023. doi: 10.1016/j.mlwa.2023.100470.

Le, V.-H. and Zhang, H. Log-based anomaly detection without log parsing. In *2021 36th IEEE/ACM International Conference on Automated Software Engineering (ASE)*, pp. 492–504, 2021. doi: 10.1109/ASE51524.2021.9678773.

Lee, K., Lee, K., Lee, H., and Shin, J. A simple unified framework for detecting out-of-distribution samples and adversarial attacks. In *Advances in Neural Information Processing Systems*, volume 31. Curran Associates, Inc., 2018.

Lipman, Y., Chen, R. T. Q., Ben-Hamu, H., Nickel, M., and Le, M. Flow matching for generative modeling. In *International Conference on Learning Representations (ICLR)*, 2023.

Luo, P., Deng, D., Xie, M., Yang, G., Chu, J., Soong, B.-H., and Yuen, C. System log anomaly detection with noise-contrastive learning and pattern feature. *IEEE Transactions on Network Science and Engineering*, 13:11–24, 2026. doi: 10.1109/TNSE.2025.3579809.

Meng, W., Liu, Y., Zhu, Y., Zhang, S., Pei, D., Liu, Y., Chen, Y., Zhang, R., Tao, S., Sun, P., and Zhou, R. Loganomaly: unsupervised detection of sequential and quantitative anomalies in unstructured logs. In *Proceedings of the 28th International Joint Conference on Artificial Intelligence*, IJCAI'19, pp. 4739–4745, Macao, China, 2019. AAAI Press.

Mylläri, J., Aalto, T., and Nurminen, J. K. Ladle: A method for unsupervised anomaly detection across log types. *Automated Software Engineering*, 32(2):34, 2025. doi: 10.1007/s10515-025-00504-w.

Narayanan, H. and Mitter, S. Sample complexity of testing the manifold hypothesis. In *Advances in Neural Information Processing Systems (NeurIPS)*, volume 23, 2010.

Ng, I., Li, Y., Li, Z., Zheng, Y., Chen, G., and Zhang, K. A general representation-based approach to multi-source domain adaptation. In *International Conference on Machine Learning (ICML)*, 2025.

Oliner, A. and Stearley, J. What supercomputers say: A study of five system logs. In *37th Annual IEEE/IFIP International Conference on Dependable Systems and Networks (DSN'07)*, pp. 575–584, 2007. doi: 10.1109/DSN.2007.103.

Saito, K., Watanabe, K., Ushiku, Y., and Harada, T. Maximum classifier discrepancy for unsupervised domain adaptation. In *IEEE/CVF Conference on Computer Vision and Pattern Recognition (CVPR)*, pp. 3723–3732, 2018.

Sharma, K. and Yelleti, V. Log anomaly detection via meta learning and prototypical networks for cross domain generalization, 2026. arXiv preprint arXiv:2601.14336.

Snell, J., Swersky, K., and Zemel, R. Prototypical Networks for Few-shot Learning. In *Advances in Neural Information Processing Systems (NeurIPS)*, volume 30. Curran Associates, Inc., 2017.

Song, Y., Sohl-Dickstein, J., Kingma, D. P., Kumar, A., Ermon, S., and Poole, B. Score-based generative modeling through stochastic differential equations. In *International Conference on Learning Representations (ICLR)*, 2021.

Sun, Y., Tao, H., and Stojanovic, V. Pseudo-label guided dual classifier domain adversarial network for unsupervised cross-domain fault diagnosis with small samples. *Advanced Engineering Informatics*, 64:102986, 2025. doi: 10.1016/j.aei.2024.102986.

Tailanian, M., Pardo, , and Musé, P. U-Flow: A U-shaped Normalizing Flow for Anomaly Detection with Unsupervised Threshold, 2024.

Vahdat, A. and Kautz, J. NVAE: A deep hierarchical variational autoencoder. In *Advances in Neural Information Processing Systems*, volume 33, pp. 19667–19679. Curran Associates, Inc., 2020.

Villani, C. *Optimal Transport: Old and New*, volume 338 of *Grundlehren der mathematischen Wissenschaften*. Springer, Berlin, Heidelberg, 2009.

Wang, M. and Deng, W. Deep visual domain adaptation: A survey. *Neurocomputing*, 312:135–153, 2018. doi: 10.1016/j.neucom.2018.05.083.

Wang, P., Qi, Y., Wang, Y., and Pan, G. Flow matching for few-trial neural adaptation with stable latent dynamics. In *International Conference on Machine Learning (ICML)*, 2025.

Wang, Z., Zhang, Z., Lee, C.-Y., Zhang, H., Sun, R., Ren, X., Su, G., Perot, V., Dy, J., and Pfister, T. Learning to prompt for continual learning. In *2022 IEEE/CVF Conference on Computer Vision and Pattern Recognition (CVPR)*, pp. 139–149, 2022. doi: 10.1109/CVPR52688.2022.00024.

Xu, W., Huang, L., Fox, A., Patterson, D., and Jordan, M. I. Detecting large-scale system problems by mining console logs. In *Proceedings of the ACM SIGOPS 22nd Symposium on Operating Systems Principles*, SOSP '09, pp. 117–132, New York, NY, USA, 2009. Association for Computing Machinery. doi: 10.1145/1629575.1629587.

Ye, X., Huang, C., Huang, H., and Yao, L. Gaussian Mixture Flow Matching with Domain Alignment for Multi-Domain Sequential Recommendation, 2025. arXiv preprint arXiv:2510.21021.

Zhao, X., Jia, T., He, M., Li, Y., and Huang, G. ZeroLog: Zero-label generalizable cross-system log-based anomaly detection, 2025.

Zhou, J., Ying, S., Wang, S., Zhao, D., Xiang, J., Liang, K., and Liu, P. LogDLR: Unsupervised cross-system log anomaly detection through domain-invariant latent representation. *IEEE Transactions on Dependable and Secure Computing*, 22(4):4456–4471, 2025. doi: 10.110 9/TDSC.2025.3548050.

Zhu, J., He, S., He, P., Liu, J., and Lyu, M. R. Loghub: A large collection of system log datasets for AI-driven log analytics. In *2023 IEEE 34th International Symposium on Software Reliability Engineering (ISSRE)*, pp. 355–366, 2023. doi: 10.1109/ISSRE59848.2023.00071.

# A. Sensitivity Analysis

In this section, we provide additional sensitivity analyses for the key hyperparameters of MPFM. Unless otherwise specified, all experiments are conducted under the cross multi-domain joint training setting, and Avg F1 denotes the unweighted arithmetic mean over HDFS, BGL, Thunderbird, and Spirit. We vary one hyperparameter at a time while keeping the remaining settings unchanged.

## A.1. Temperature and Hard Sample Count

We first analyze the temperature parameter $\tau$ used in prototype assignment and contrastive learning, and the number of selected hard samples $K$. The results are shown in Table 6.

*Table 6.* Sensitivity analysis of the temperature parameter $\tau$ and the number of hard samples $K$.

| $\tau$ | 0.05 | 0.20 | 0.50 | 1.00 | 5.00 |
|---|---|---|---|---|---|
| Avg F1 | 0.917 | 0.964 | 0.960 | 0.943 | 0.882 |

| $K$ | 8 | 16 | 32 | 64 | 128 |
|---|---|---|---|---|---|
| Avg F1 | 0.946 | 0.957 | 0.960 | 0.962 | 0.931 |

For $\tau$, a very small value such as $0.05$ makes the prototype assignment overly sharp and reduces the model's ability to cover multimodal normal patterns. A very large value such as $5.00$ makes the assignment too smooth, weakening the discriminative effect of prototypes. Although $\tau = 0.20$ yields a slightly higher Avg F1, it leads to less stable training on noisier datasets in our experiments. Therefore, we use $\tau = 0.50$ as a more robust default setting.

For the hard sample count $K$, MPFM is stable within a moderate range. Increasing $K$ from 32 to 64 brings only a marginal improvement, while $K = 128$ significantly degrades performance. This suggests that selecting too many hard candidates may introduce noisy or less informative samples into the hard mining process. We therefore set $K = 32$ as the default configuration to balance performance and efficiency.

## A.2. Loss Weight Sensitivity

We then study the sensitivity of the loss weights $\lambda_1$, $\lambda_2$, and $\lambda_3$. Here, $\lambda_1$ controls the InfoNCE loss, $\lambda_2$ controls the orthogonal regularization, and $\lambda_3$ controls the hard flow loss.

*Table 7.* Sensitivity analysis of $\lambda_1$ and $\lambda_2$.

| Weight | 0.1 | 0.5 | 1.0 | 2.0 | 5.0 |
|---|---|---|---|---|---|
| $\lambda_1$ (InfoNCE) | 0.895 | 0.963 | 0.960 | 0.951 | 0.912 |
| $\lambda_2$ (Ortho) | 0.911 | 0.956 | 0.960 | 0.961 | 0.934 |

As shown in Table 7, MPFM is relatively stable around the default setting $\lambda_1 = \lambda_2 = 1.0$. For $\lambda_1$, a very small weight weakens the boundary-separating effect of contrastive learning, while an overly large weight may overemphasize contrastive separation and reduce the stability of prototype representation learning. For $\lambda_2$, moderate orthogonal regularization helps prevent prototype collapse, while excessive regularization may overly constrain prototype adaptation. Overall, both losses show broad stability around the default values.

*Table 8.* Sensitivity analysis of the hard flow loss weight $\lambda_3$.

| $\lambda_3$ | 0.5 | 1.0 | 5.0 | 10.0 |
|---|---|---|---|---|
| Avg F1 | 0.957 | 0.960 | 0.925 | 0.885 |
| Avg Precision | 0.952 | 0.958 | 0.887 | 0.814 |
| Avg Recall | 0.962 | 0.962 | 0.967 | 0.969 |

Table 8 shows that excessively increasing $\lambda_3$ leads to a clear precision drop, while recall only increases slightly. This indicates that an overly large hard mining weight makes the model too sensitive near decision boundaries and causes it to

misclassify more normal samples as anomalies. In other words, excessive hard mining may lead to boundary overfitting. This observation is consistent with the ablation study, where hard mining acts as an auxiliary component for boundary robustness rather than the dominant source of performance gain. Therefore, we use $\lambda_3 = 1.0$ as the default setting.

### A.3. Prototype Drift Extrapolation

The extrapolation coefficient $\beta$ controls how far hard candidates are generated along the prototype drift direction. The results are shown in Table 9.

*Table 9.* Sensitivity analysis of the prototype drift extrapolation coefficient $\beta$.

| $\beta$ | 0.1 | 0.3 | 0.5 | 0.7 | 0.9 |
|---|---|---|---|---|---|
| Avg F1 | 0.948 | 0.956 | 0.960 | 0.951 | 0.931 |

The model performs best around $\beta = 0.5$ and remains stable in the range of 0.3 to 0.7. When $\beta$ is too small, the generated candidates remain too close to the current prototypes and provide limited boundary information. When $\beta$ is too large, the generated candidates may move beyond the drift-induced uncertainty region and become less informative or even noisy. Therefore, $\beta = 0.5$ provides a practical balance between boundary exploration and training stability.

### A.4. Number of Shared and Private Prototypes

We further analyze the influence of the numbers of shared prototypes $K_s$ and private prototypes $K_p$. The results are reported in Table 10.

*Table 10.* Sensitivity analysis of the numbers of shared and private prototypes.

| $K_s$ | $K_p$ | HDFS | BGL | TB | Spirit | Avg F1 |
|---|---|---|---|---|---|---|
| 16 | 8 | 0.979 | 0.946 | 0.903 | 0.963 | 0.948 |
| 32 | 16 | 0.987 | 0.959 | 0.915 | 0.977 | 0.960 |
| 64 | 32 | 0.982 | 0.952 | 0.909 | 0.970 | 0.953 |
| 32 | 0 | 0.968 | 0.937 | 0.891 | 0.941 | 0.934 |

The default setting $K_s = 32$ and $K_p = 16$ achieves the best overall performance. Using fewer prototypes reduces the model's ability to represent diverse normal patterns, while using more prototypes does not bring further improvement and may introduce redundant prototype assignments. Notably, removing private prototypes by setting $K_p = 0$ causes the largest performance drop, reducing Avg F1 from 0.960 to 0.934. This confirms the importance of preserving domain-specific normal structures in the cross multi-domain joint training setting.

### A.5. Inference Time Point for Flow Error

During inference, MPFM evaluates the flow matching error at a fixed time point $t^*$. We analyze the sensitivity of this choice in Table 11.

*Table 11.* Sensitivity analysis of the inference time point $t^*$.

| $t^*$ | 0.1 | 0.3 | 0.5 | 0.7 | 0.9 |
|---|---|---|---|---|---|
| HDFS | 0.982 | 0.986 | 0.987 | 0.985 | 0.979 |
| BGL | 0.938 | 0.952 | 0.959 | 0.951 | 0.934 |
| TB | 0.889 | 0.905 | 0.915 | 0.909 | 0.884 |
| Spirit | 0.957 | 0.970 | 0.977 | 0.968 | 0.949 |
| Avg F1 | 0.942 | 0.953 | 0.960 | 0.953 | 0.937 |

The performance is stable when $t^*$ lies in the middle region of the flow trajectory. In particular, the Avg F1 varies only slightly between $t^* = 0.3$ and $t^* = 0.7$. When $t^*$ is too small, the intermediate state is closer to the noise distribution, making the flow error less discriminative. When $t^*$ is too large, the state approaches the target endpoint, and the velocity residuals become uniformly smaller, compressing the difference between normal and anomalous samples. Therefore, we use $t^* = 0.5$ as a robust default choice.

Overall, MPFM shows stable performance across a reasonably wide range of hyperparameter settings. The most influential design is the use of private prototypes, whose removal leads to the largest performance degradation. Flow-related parameters such as $\beta$, $\lambda_3$, and $t^*$ also affect the boundary behavior of the model, but moderate default values provide stable performance. These results indicate that MPFM does not rely on a narrow hyperparameter configuration and can be practically tuned under the cross multi-domain joint training setting.

## B. Multi-Seed Stability

To evaluate the robustness of MPFM with respect to random initialization, we repeat MPFM and the strongest joint-training baseline LogDLR with five random seeds under the cross multi-domain joint training setting. The results are reported as mean $\pm$ standard deviation in Table 12.

*Table 12.* Multi-seed stability comparison under the cross multi-domain joint training setting. We report F1 scores as mean $\pm$ standard deviation over five random seeds.

| Method | HDFS | BGL | TB | Spirit | Avg F1 |
|---|---|---|---|---|---|
| LogDLR | 0.848±0.014 | 0.809±0.018 | 0.772±0.023 | 0.876±0.016 | 0.826±0.015 |
| MPFM | 0.984±0.004 | 0.955±0.008 | 0.909±0.012 | 0.973±0.007 | 0.955±0.006 |

As shown in Table 12, MPFM consistently outperforms LogDLR across all four datasets and achieves smaller variance under different random seeds. The improvement margin is substantially larger than the observed standard deviation, indicating that the performance gain is not caused by a favorable initialization. These results further confirm the stability of MPFM under the cross multi-domain joint training setting.

## C. Additional Baseline Comparisons

In addition to the main baselines reported in the main text, we further compare MPFM with recent language-model-based, zero-shot, and meta-learning log anomaly detection methods. Since these methods follow different training and evaluation protocols, we report them as contextual comparisons rather than strictly same-protocol comparisons. In particular, LogFiT, LogGPT, Baichuan, and ChatGLM2 are evaluated in single-domain or few-shot settings, while ZeroLog and MetaLog follow pairwise transfer or target-domain adaptation protocols. In contrast, MPFM is trained once on pooled multi-domain data and deployed as a single model across all domains.

*Table 13.* Additional comparison with language-model-based, zero-shot, and meta-learning baselines. "–" indicates that the corresponding result is not reported by the original method. These results are contextual comparisons because the compared methods follow different training protocols.

| Method | Type | HDFS | BGL | TB | Spirit |
|---|---|---|---|---|---|
| LogFiT | Encoder LM (single-domain) | 0.950 | 0.899 | 0.939 | – |
| LogGPT | Prompt-based LLM (few-shot) | – | 0.625 | – | 0.740 |
| Baichuan | General LLM (few-shot) | – | 0.325 | – | 0.527 |
| ChatGLM2 | General LLM (few-shot) | – | 0.201 | – | 0.137 |
| ZeroLog | Zero-shot transfer (pairwise) | 0.806 | 0.860 | – | – |
| MetaLog | Meta-learning (1% target labels) | 0.815 | 0.929 | 0.320 | – |
| MPFM | Multi-domain joint training | 0.987 | 0.959 | 0.915 | 0.977 |

As shown in Table 13, MPFM achieves strong performance compared with recent language-model-based methods. Although language models are effective for textual pattern modeling, purely text-based modeling does not necessarily provide sufficient cross-domain anomaly detection capability, especially when the model needs to handle heterogeneous systems with a unified detector. MPFM outperforms the LLM-based baselines on most comparable datasets, suggesting that structural prototype modeling and flow-based dynamical consistency provide useful inductive biases beyond text representation learning.

Compared with ZeroLog and MetaLog, MPFM also differs in deployment protocol. ZeroLog and MetaLog are designed for pairwise transfer or target-domain adaptation, where each new target domain may require a separate adaptation procedure. MetaLog also uses a small portion of target-domain labels. In contrast, MPFM follows a single-model multi-domain deployment setting: one model is jointly trained on multiple domains and then evaluated on each domain without per-target

re-adaptation. Under this setting, MPFM achieves higher scores on the comparable datasets and provides a more scalable solution for multi-system log anomaly detection.

## D. Model Complexity and Additional Efficiency Details

In addition to the inference efficiency comparison reported in the main text, we provide the model complexity comparison in Table 14. All experiments are conducted on a single RTX 3090. The reported training time corresponds to 30 training epochs.

*Table 14.* Model complexity comparison. Inference memory denotes the GPU memory usage during inference. Training time is measured on a single RTX 3090 for 30 epochs.

| Method | Params (M) | Infer. Mem. (GB) | Train Time (h) |
|--------|-----------|------------------|----------------|
| LogBERT | 12.5 | 1.9 | $\sim$4.2 |
| LogDLR | 6.3 | 1.6 | $\sim$5.8 |
| MPFM | 3.8 | 2.3 | $\sim$7.5 |

As shown in Table 14, MPFM has fewer parameters than LogBERT and LogDLR, mainly because it uses a compact Transformer encoder together with prototype and flow modules rather than a large language-model-style encoder. However, MPFM requires slightly higher inference memory than the baselines. This overhead mainly comes from the ODE-based flow evaluation, where intermediate states along the trajectory need to be maintained during inference.

The training time of MPFM is also longer than that of LogBERT and LogDLR. The additional cost mainly comes from prototype-drift-driven hard example mining, which requires extra flow-error computation for candidate hard samples. It is worth noting that the training-time flow matching objective only requires single-step forward evaluation for sampled time points, rather than multi-step ODE integration. Multi-step ODE solving is mainly used during inference when computing the final flow-based anomaly score.

Together with the inference efficiency results in the main text, these observations indicate that MPFM introduces a controllable computational overhead. In particular, reducing the number of ODE steps provides a practical way to trade accuracy for efficiency. The 5-step setting achieves a substantially higher throughput than the 20-step setting with only a moderate decrease in Avg F1, making it a practical configuration for deployment scenarios where latency is more important.

