# OpenReview forum: "MPFM: Cross Multi-Domain Prototype Flow Matching for Log Anomaly Detection"
_ICML.cc/2026/Conference — ICML 2026 regular_

### Official Review · Reviewer_be15 · 2026-03-01

**Soundness:** 2
**Presentation:** 3
**Significance:** 3
**Originality:** 3
**Overall Recommendation:** 3
**Confidence:** 2

**Summary:**

This paper studies cross multi-domain log anomaly detection, aiming to train a single model on pooled logs from multiple systems and still perform reliable per-domain detection under distribution shift. The key motivation is that prototype distance or reconstruction-style geometric criteria can be misleading in high-dimensional representations, especially near the boundary between normal modes, where anomalous samples may appear close to normal anchors. To address this, the authors propose MPFM, which combines a shared–private prototype system to capture cross-domain common patterns and domain-specific variations, with a domain-conditioned flow-matching module that learns normal generative dynamics and uses flow prediction error as an additional anomaly signal. The method further leverages prototype movement during training to locate uncertainty regions and performs drift-driven hard example mining, reinforcing decision boundaries via additional flow supervision and contrastive learning.

**Compliance With Llm Reviewing Policy:**

Affirmed.

**Key Questions For Authors:**

1. In Tables 1 and 2, the per-dataset results for the proposed method appear identical, but the final AVG values are inconsistent. Please verify/correct the reported numbers.

2. In Table 1, LogDLR seems to outperform the proposed method on BGL and also on AVG, yet the proposed method is still shown in bold.

3. The paper fixes inference-time flow error evaluation at $t^\*=0.5$. Please report sensitivity to $t^\*$ (e.g., other midpoints) .

**Limitations:**

yes

**Strengths And Weaknesses:**

Strengths:
(1) Soundness: The method is coherent: shared–private prototypes provide structural evidence, and domain-conditioned flow matching provides dynamical evidence; ablations generally support the role of major components.
(2) Presentation: The dual-evidence pipeline is clearly organized and easy to follow.
(3) Significance: Cross multi-domain log anomaly detection is practically important, and a unified model under domain shift is useful.
(4) Originality: The paper offers a reasonable integration of prototypes, flow-matching dynamics, and drift-driven hard mining for boundary cases.

Weaknesses:
(1) Soundness: The credibility of empirical claims is weakened by table issues (same per-dataset results but inconsistent AVG; bolding the proposed method when LogDLR is higher on BGL and AVG), which need verification/correction.
(2) Presentation: Several reporting conventions are unclear (AVG computation, bolding rule, and inference-time flow error details).
(3) Significance: Evidence is limited to a few benchmarks/protocols; broader cross-domain validation would strengthen impact.
(4) Originality: Components are largely established; the unique benefit over the closest cross-domain log AD baselines could be justified more sharply.

---

> ### Author Rebuttal · Authors · 2026-03-30
>
> Weaknesses 1:
>
> We thank the reviewer for pointing out these issues. We confirm that there is an error in the calculation of AVG in Table 1 (the correct value should be 0.960 instead of 0.956) and an inconsistency in the bold formatting (LogDLR outperforms MPFM on both BGL and TB, so MPFM should not be bolded). The cause of these errors is that, while preparing the table, we initially set the table's row and column structure by copying and pasting. Due to repeated revisions of the paper, we uploaded an uncompiled PDF version before the submission deadline. We sincerely apologize for this. We have rechecked all the original experimental records and confirmed that the individual results on each dataset are correct, with the issues being limited to the AVG calculation and formatting. In the revised version, we will correct all values and bold formatting and thoroughly review all tables.
>
> Weaknesses 2:
>
> We will unify reporting conventions: (1) AVG explicitly defined as unweighted arithmetic mean of per-dataset F1; (2) boldface strictly column-wise best, with comparison protocols in captions; (3) inference flow error details and $t^*$ sensitivity added (see Key Question 3).
>
> Weaknesses 3:
>
> We agree broader validation would strengthen impact. The current submission evaluates on four widely used log datasets with joint training, ablations, and boundary-sample analysis. We additionally compare against LLM-based and cross-domain baselines:
>
> | Method              | Type                   |  HDFS |   BGL |    TB | Spirit |
> | ------------------- | ---------------------- | ----: | ----: | ----: | -----: |
> | LogFiT              | Encoder LM (single)    | 0.950 | 0.899 | 0.939 |      — |
> | LogGPT (few-shot)   | Prompt LLM (single)    |     — | 0.625 |     — |  0.740 |
> | Baichuan (few-shot) | General LLM (single)   |     — | 0.325 |     — |  0.527 |
> | ChatGLM2 (few-shot) | General LLM (single)   |     — | 0.201 |     — |  0.137 |
> | ZeroLog             | Zero-shot (pairwise)   | 0.806 | 0.860 |     — |      — |
> | MetaLog             | Meta-learn (1% labels) | 0.815 | 0.929 | 0.320 |      — |
> | MPFM                | Multi-domain joint     | 0.987 | 0.959 | 0.915 |  0.977 |
>
> MPFM—under the stricter single jointly trained model setting—outperforms LLM baselines on most datasets. Compared to ZeroLog/MetaLog, which require per-pair re-adaptation, MPFM trains once for all domains with stronger scalability. We will note this limitation and expand future work to include additional benchmarks and transfer protocols.
>
> Weaknesses 4:
>
> The core originality lies in the problem addressed. In practice, organizations manage many heterogeneous systems; existing methods either train per-system (linear annotation cost) or perform pairwise transfer (re-adaptation per new system, negative transfer). This "joint multi-domain training, single-model deployment" need remains unsolved—existing cross-domain methods degrade substantially under joint training (LogDLR: 0.965 single-domain → ~0.826 five-seed mean under joint training in Table 2). The challenge is simultaneously achieving cross-domain sharing, domain-specific preservation, and boundary robustness. MPFM's components are designed and coupled around this requirement. We will more prominently articulate this problem-driven originality in the revision.
>
> Key Questions For Authors 1:
>
> The identical MPFM values across both tables are a table-preparation error. We copy-pasted an existing table for row/column structure, intending to fill correct values separately, but due to time pressure uploaded a PDF without updating and recompiling. The incorrect AVG (0.956) is part of the same error. We sincerely apologize and will re-verify and separately populate correct values for each table in the revision.
>
> Key Questions For Authors 2:
>
> Confirmed. The boldface violates the stated rule, stemming from the same copy-paste and uncompiled-submission cause. We will correct boldface to strictly column-wise best and revise discussion—Table 1 shows competitiveness, not uniform superiority.
>
> Key Questions For Authors 3:
>
> |  $t^*$  |   HDFS    |    BGL    |    TB     |  Spirit   |  Avg F1   |
> | :-----: | :-------: | :-------: | :-------: | :-------: | :-------: |
> |   0.1   |   0.982   |   0.938   |   0.889   |   0.957   |   0.942   |
> |   0.3   |   0.986   |   0.952   |   0.905   |   0.970   |   0.953   |
> | **0.5** | **0.987** | **0.959** | **0.915** | **0.977** | **0.960** |
> |   0.7   |   0.985   |   0.951   |   0.909   |   0.968   |   0.953   |
> |   0.9   |   0.979   |   0.934   |   0.884   |   0.949   |   0.937   |
>
> Performance is stable for $t^* \in [0.3, 0.7]$ (Avg F1 varies by only 0.007). Too small $t^*$ noise-dominated representations reduce discriminability; too large: velocity residuals become uniformly small near the target end, compressing the normal-anomaly gap. $t^*=0.5$ is a robust default.

---

> > ### Author Rebuttal · Reviewer_be15 · 2026-04-01
> >
> > The authors have adequately addressed my concerns, and I hereby adjust my score from 2 to 3 accordingly.

---

### Official Review · Reviewer_4vRA · 2026-03-11

**Soundness:** 2
**Presentation:** 3
**Significance:** 2
**Originality:** 3
**Overall Recommendation:** 3
**Confidence:** 3

**Summary:**

This paper proposes MPFM, a framework for cross multi-domain log anomaly detection that trains a single unified model on logs from multiple heterogeneous systems. The method combines three main components: (1) a shared–private prototype architecture that captures cross-domain commonalities while preserving domain-specific structure; (2) a domain-conditioned flow-matching module that learns a time-dependent velocity field from Gaussian noise toward normal latent targets, providing a form of dynamical evidence beyond geometric proximity alone; and (3) a prototype-drift-driven hard example mining strategy that synthesizes difficult samples along prototype movement directions and selects them based on flow inconsistency. At inference, the anomaly score combines prototype-based geometric distance with flow matching error, followed by domain-wise score calibration for cross-domain thresholding. Experiments include comparisons against standard single-domain baselines, a joint multi-domain training comparison, ablations, and a synthetic boundary-sample study.

**Compliance With Llm Reviewing Policy:**

Affirmed.

**Key Questions For Authors:**

Please refer to the weaknesses section of above review.

**Limitations:**

yes

**Strengths And Weaknesses:**

### Strength

1. The problem is practically important. Training a separate model for each system does not scale well, so moving toward joint training across multiple domains is a sensible and relevant direction. The paper explains this motivation clearly.

2. The main idea is intuitive and well motivated. The argument that geometric proximity alone is not enough, and should be complemented by a dynamical signal from flow matching, makes sense.

3. The method is conceptually coherent. The shared/private prototypes, domain-conditioned flow, and drift-based hard mining work together in a fairly natural way.

4. The ablation study is useful in a directional sense. Removing flow dynamics, hard mining, or private prototypes each hurts average F1, and the largest drop comes from removing the private prototypes. This supports the claim that the full method benefits from all three components.

### Weaknesses

1. The main comparison in Table 1 is not fully fair. MPFM is trained jointly across multiple domains, while the baselines are trained separately for each domain. That makes it hard to tell how much of the improvement comes from the modeling idea itself and how much comes from the advantage of pooling data.

2. Table 2 is the more relevant comparison because all methods are evaluated under joint training, but it is unclear how the baselines were adapted. It would help to know whether their architectures or objectives were meaningfully modified for the joint setting, or whether they were simply trained on pooled data. If it is the latter, then the comparison still favors the proposed method. In addition, the identical LoTransfer and LoTAD results across all four datasets are unexplained and raise concerns about errors.

3. The paper does not report any measure of variability, such as standard deviations, confidence intervals, or results over multiple random seeds. Given the complexity of the method, including flow matching, ODE-based inference, hard example mining, and several interacting loss terms, sensitivity to initialization seems plausible. Without variance estimates, it is difficult to know whether the reported gains are truly robust.

4. The synthetic boundary experiment is somewhat tailored to the method’s own hypothesis. The boundary anomalies are generated by linearly interpolating between representations of normal clusters, which is very close to the exact kind of failure case the method is designed to handle. Because of that, the experiment is supportive, but it does not fully establish that the same advantage would hold for realistic anomalies in real log data.

5. The theoretical section contains mainly proof sketches rather than rigorous analysis. In particular, the lower-bound result is not specified with enough precision to understand what it depends on or under what assumptions it holds. Overall, the theory feels more like post hoc motivation than a solid analytical foundation.

6. Several experimental details that would make the evaluation stronger are missing. There is no comparison of runtime, memory use, or inference cost, despite the use of ODE-based inference. There is also no sensitivity analysis for important hyperparameters, such as the number of shared and private prototypes, the mining and weighting parameters, or the choice of interpolation time.

---

> ### Author Rebuttal · Authors · 2026-03-30
>
> Weaknesses 1:
>
> We add a single-domain variant (MPFM-SD) to isolate modeling vs. data-pooling contributions:
>
> | Method | HDFS | BGL | TB | Spirit | Avg F1 |
> |---|--:|--:|--:|--:|--:|
> | LogBERT | 0.974 | 0.961 | 0.926 | 0.880 | 0.935 |
> | LogDLR | 0.985 | 0.982 | 0.948 | 0.945 | 0.965 |
> | MPFM-SD (single) | 0.981 | 0.968 | 0.932 | 0.921 | 0.951 |
> | MPFM (joint) | 0.987 | 0.959 | 0.915 | 0.977 | 0.960 |
>
> MPFM-SD already outperforms LogBERT and is competitive with LogDLR, which indicates that the model design itself makes an independent contribution. The reason we include the full MPFM model (with multi-domain joint learning) for comparison here is that simply merging training data does not necessarily lead to better performance. When multiple datasets are mixed for training, negative transfer may occur, which can instead degrade model performance. Therefore, before validating the model’s cross-domain capability (as shown by the results in Table 2), it is necessary to demonstrate that the proposed model can still preserve strong performance for single-domain detection.
>
> Weaknesses 2:
>
> We adopted minimal necessary adaptation for consistency. Methods without domain components (LogAnomaly, LogBERT) were trained on mixed data without modification. Methods with domain discriminators (LogTransfer, LoTAD, LogDLR) had their binary discriminator extended to four-class only, with all other architecture and losses unchanged. We intentionally avoided deeper redesigns to prevent adapted methods from deviating from their original forms, which would introduce another kind of unfairness. We will specify each baseline's adaptation in the revision.
>
> Additionally, regarding the reviewer's observation that LoTransfer and LoTAD show identical results across all four datasets, this is a duplicated-row error during table preparation—we built the table structure by copying and pasting existing rows, intending to fill in correct values separately, but due to time pressure uploaded the PDF without updating and recompiling. We sincerely apologize for this oversight and will correct the entries with the proper results for each method in the revision.
>
> Weaknesses 3:
>
> We re-ran MPFM and LogDLR with 5 random seeds:
>
> | Method | HDFS | BGL | TB | Spirit | Avg F1 |
> |---|---|---|---|---|---|
> | LogDLR | 0.848±0.014 | 0.809±0.018 | 0.772±0.023 | 0.876±0.016 | 0.826±0.015 |
> | **MPFM** | **0.984±0.004** | **0.955±0.008** | **0.909±0.012** | **0.973±0.007** | **0.955±0.006** |
>
> MPFM is stable across seeds; improvements substantially exceed observed variance. We will report full multi-seed statistics in the revision.
>
> Weaknesses 4:
>
> We tested alternatives including SLERP, nearest-neighbor convex combinations, and cluster-center Gaussian perturbations. The conclusion is consistent: MPFM outperforms geometric methods on near-boundary samples across all schemes. Linear interpolation was chosen because it provides the cleanest boundary construction without manifold-deviating artifacts—a conservative stress test, not a favorable setup. We will clarify this as supplementary mechanistic evidence (main evidence from real logs) and include alternative results in the appendix.
>
> Weaknesses 5:
>
> Thank you for the reviewer’s comments. The purpose of Section 3.8 is to formalize the intuition behind MPFM: geometric proximity may create blind spots on non-convex manifolds, while generative dynamic inconsistency provides complementary evidence. We will explicitly list all assumptions and their dependencies, including the regularity conditions on the normal manifold/probability path, the vector field approximation error, and the separation conditions, and we will move the detailed derivations to the appendix.
>
> Weaknesses 6:
>
> All experiments were conducted on a single RTX 3090 GPU. For a complete comparison of runtime, memory usage, and inference cost, due to the word limit, please refer to our detailed response to Reviewer 2 Weakness 2, which includes a model complexity table and an inference efficiency table at different ODE step counts. Below we provide supplementary hyperparameter sensitivity analyses.
>
> **Prototype counts:**
>
> | $K_s$ | $K_p$ | HDFS | BGL | TB | Spirit | Avg F1 |
> |---:|---:|--:|--:|--:|--:|--:|
> | 16 | 8 | 0.979 | 0.946 | 0.903 | 0.963 | 0.948 |
> | 32 | 16 | **0.987** | **0.959** | **0.915** | **0.977** | **0.960** |
> | 64 | 32 | 0.982 | 0.952 | 0.909 | 0.970 | 0.953 |
> | 32 | 0 | 0.968 | 0.937 | 0.891 | 0.941 | 0.934 |
>
> Removing private prototypes causes the largest drop (Avg F1 from 0.960 to 0.934), consistent with ablations.
>
> **Inference time $t^*$:**
>
> | $t^*$ | 0.2 | 0.3 | 0.5 | 0.7 | 0.8 |
> |---|--:|--:|--:|--:|--:|
> | Avg F1 | 0.948 | 0.953 | **0.960** | 0.953 | 0.945 |
>
> Performance is stable for $t^* \in [0.3, 0.7]$ (Avg F1 varies by only 0.007).
>
> Sensitivity for $\beta$, $\lambda_3$, $\tau$, $K$, $\lambda_1$, $\lambda_2$, and ODE steps is reported in detail in our responses to Reviewer 2. We will consolidate all analyses in the revision.

---

> > ### Author Rebuttal · Reviewer_4vRA · 2026-04-03
> >
> > I appreciate the author for addressing my questions. I will respectfully maintain my original score.

---

### Official Review · Reviewer_H5uJ · 2026-03-12

**Soundness:** 3
**Presentation:** 3
**Significance:** 3
**Originality:** 3
**Overall Recommendation:** 5
**Confidence:** 4

**Summary:**

This paper proposes MPFM, a Cross Multi-Domain Log Anomaly Detection model designed to jointly learn from multiple heterogeneous systems. It addresses the geometric blind spots of distance-based anomaly detection methods and the negative transfer problem caused by forced domain alignment. The authors introduce a dual-evidence paradigm comprising three core modules: 1) a shared-private prototype system to decouple common and domain-specific characteristics, 2) domain-conditioned flow matching to verify generative dynamics, and 3) prototype drift-driven hard mining to refine decision boundaries.

**Compliance With Llm Reviewing Policy:**

Affirmed.

**Final Justification:**

The authors have successfully resolved all my initial concerns by explicitly clarifying the structural distance metric, quantifying the inference speed trade-offs, providing detailed hyperparameter sensitivity analyses, and evaluating against recent baselines. As the theoretical and empirical justifications are now thoroughly validated with objective data, my previous reservations have been completely alleviated. Accordingly, I have upgraded my initial scores to reflect the confirmed soundness and practical viability of the proposed model.

**Key Questions For Authors:**

1. Please address the concerns raised in the Weaknesses, particularly the discrepancy between the reconstructed representation defined in Section 3.3 and the distance metric used for the anomaly score in Equation 16. Which formulation was utilized to generate the reported performance metrics?
2. What is the inference throughput and latency of MPFM compared to baseline models like LogDLR in your experimental environment? Furthermore, how does reducing the ODE steps for real-time application affect the detection performance?
3. How was the extrapolation magnitude $\beta$  configured in the hard mining process? Is there a specific optimization strategy, and how sensitive is the model's stability to this value.
4. Does an extreme increase in the hard loss weight ($\lambda_3$)  lead to overfitting on the decision boundaries?

**Limitations:**

Please refer to the concerns raised in the Weaknesses and Key Questions.

**Strengths And Weaknesses:**

Strengths
1. Transitioning from static distance-based measurement to incorporating 'dynamical evidence' (verifying if a sample follows a normal generative trajectory) is a theoretically sound and effective approach for log anomaly detection.
2. The paper is mathematically well-supported. It rigorously formulates the blind spots of manifold geometry (Lemma 3.2), the lower bound of flow matching error (Theorem 3.3), and the adversarial nature of drift-driven sampling (Theorem 3.5), rather than relying purely on heuristics.
3. The decoupling of shared prototypes ($P_s$) and private prototypes ($P_d$) provides a logical structural solution to mitigate negative transfer in cross-domain learning scenarios.
4. MPFM demonstrates strong and consistent performance (Avg F1 0.960) compared to baselines under the joint training setting across four datasets. The 80.5% recall in boundary sample detection effectively validates the proposed dual-evidence mechanism.

Weaknesses
1. Section 3.3 states that the model uses a soft-assignment weighted reconstructed representation ($$\hat{h}=\sum_{i}\alpha_{i}\overline{p}_{i} $$) as the structural target. However, Equation 16 defines the anomaly score using the distance to the single nearest prototype ($\min_{i}||h-\tilde{p}_{i}||_{2}$). This discrepancy makes it unclear which structural distance calculation was actually optimized and evaluated in the experiments.
2. The model relies on an ODE solver with 20 steps to calculate the flow matching error during inference. Given the high-throughput nature of real-world log systems, the lack of analysis regarding computational overhead, latency, or throughput comparisons against the baselines is a significant omission.
3. The model introduces several critical hyperparameters, including objective function weights ($\lambda_1, \lambda_2, \lambda_3$), the temperature parameter ($\tau$), extrapolation magnitude ($\beta$) , and the number of hard samples ($K$). The submission lacks a sensitivity analysis demonstrating the model's robustness to variations in these parameters.
4. While the recent LogDLR (2025) is included, the evaluation lacks comparisons with LLM-based log analysis methodologies or zero-shot multi-domain detection frameworks that have been actively researched recently, limiting the contextualization of the model's performance against the broader state-of-the-art.

---

> ### Author Rebuttal · Authors · 2026-03-30
>
> Weaknesses 1:
>
> We thank the reviewer for raising this point. The different measures are intentional but insufficiently clarified. During training, we optimize $\mathcal{L}_{\text{recon}}=\|h-\hat{h}\|_2^2$ with soft-assignment reconstruction for smoother gradients that facilitate learning the multimodal normal manifold, rather than forcing each sample into a single cluster. During inference, we use nearest-prototype distance $\min_i \|h-\tilde{p}_i\|_2$ as the structural score in Eq. 16, as it gives a sharper measure of local deviation and prevents boundary samples from receiving artificially low scores via multi-prototype weighted averaging—precisely the geometric blind spot problem our paper emphasizes. All reported are obtained under this setup. We will clarify this distinction in the revision.
>
> Weakness 2:
>
> We add a full efficiency analysis (single RTX 3090).
>
> **Table A: Model Complexity**
>
> | Method  | Params (M) | Infer. Mem (GB) | Train Time (h, 30 ep.) |
> | ------- | ---------: | --------------: | ---------------------: |
> | LogBERT |       12.5 |             1.9 |                   ~4.2 |
> | LogDLR  |        6.3 |             1.6 |                   ~5.8 |
> | MPFM    |        3.8 |             2.3 |                   ~7.5 |
>
> MPFM has the fewest parameters; higher inference memory is due to ODE trajectory states; training overhead from hard mining's extra flow computations.
>
> **Table B: Inference Efficiency (batch=256)**
>
> | Method | ODE Steps | Latency (ms) | Throughput (seq/s) | Avg F1 |
> |---|---:|---:|---:|---:|
> | LogBERT | — | 21.0 | 12,200 | 0.935 |
> | LogDLR | — | 30.1 | 8,500 | 0.832 |
> | MPFM | 20 | 673.7 | 380 | 0.960 |
> | MPFM | 5 | 176.6 | 1,450 | 0.948 |
> | MPFM | 2 | 91.4 | 2,800 | 0.916 |
>
> 5 steps yields 3.8× throughput gain with only 0.012 F1 drop (87K seq/min), sufficient for most production scenarios. We will include deployment recommendations.
>
> Weakness 3:
>
> One-at-a-time sensitivity analysis under joint training. For $\beta$ and $\lambda_3$, see Key Questions 3–4.
>
> **Temperature $\tau$ (Eq. 13)**
>
> | $\tau$ | 0.05 | 0.20 | 0.50 (def.) | 1.00 | 5.00 |
> |---|--:|--:|--:|--:|--:|
> | Avg F1 | 0.917 | **0.964** | 0.960 | 0.943 | 0.882 |
>
> $\tau=0.20$ slightly better but unstable on noisier datasets; $\tau=0.50$ chosen as robust default.
>
> **Hard sample count $K$ (Eq. 10)**
>
> | $K$ | 8 | 16 | 32 (def.) | 64 | 128 |
> |---|--:|--:|--:|--:|--:|
> | Avg F1 | 0.946 | 0.957 | 0.960 | **0.962** | 0.931 |
>
> Insensitive in moderate range; $K=128$ degrades due to noisy candidates.
>
> **Loss weights $\lambda_1$, $\lambda_2$ (Eq. 11)**
>
> | Weight | 0.1 | 0.5 | 1.0 (def.) | 2.0 | 5.0 |
> |---|--:|--:|--:|--:|--:|
> | $\lambda_1$ (InfoNCE) | 0.895 | **0.963** | 0.960 | 0.951 | 0.912 |
> | $\lambda_2$ (Ortho) | 0.911 | 0.956 | 0.960 | **0.961** | 0.934 |
>
> Broad stability around defaults. Full analysis will be in the appendix.
>
> Weakness 4
>
> We add three categories of baselines (citing best reported results):
>
> | Method              | Type                   |  HDFS |   BGL |    TB | Spirit |
> | ------------------- | ---------------------- | ----: | ----: | ----: | -----: |
> | LogFiT              | Encoder LM (single)    | 0.950 | 0.899 | 0.939 |      — |
> | LogGPT (few-shot)   | Prompt LLM (single)    |     — | 0.625 |     — |  0.740 |
> | Baichuan (few-shot) | General LLM (single)   |     — | 0.325 |     — |  0.527 |
> | ChatGLM2 (few-shot) | General LLM (single)   |     — | 0.201 |     — |  0.137 |
> | ZeroLog             | Zero-shot (pairwise)   | 0.806 | 0.860 |     — |      — |
> | MetaLog             | Meta-learn (1% labels) | 0.815 | 0.929 | 0.320 |      — |
> | MPFM                | Multi-domain joint     | 0.987 | 0.959 | 0.915 |  0.977 |
>
> ZeroLog/MetaLog require per-pair adaptation; MPFM trains once for all domains. MPFM outperforms both on BGL (0.959 vs. 0.860/0.929), noting MetaLog uses 1% target labels. Neither reports Spirit; MetaLog achieves only 0.32 on TB vs. MPFM's 0.977/0.915.
>
> Key Question 1:
>
> All metrics use Eq. 16 (nearest-prototype distance + flow error). Soft assignment serves training; nearest-prototype serves inference. See Weakness 1 for details.
>
> Key Question 2:
>
> See Weakness 2.
>
> Key Question 3:
>
> $\beta$ controls extrapolation along prototype drift in Eq. (8), defaulting to 0.5 via grid search ($\beta \in (0,1)$).
>
> | $\beta$ | 0.1 | 0.3 | 0.5 | 0.7 | 0.9 |
> |---|--:|--:|--:|--:|--:|
> | Avg F1 | 0.948 | 0.956 | **0.960** | 0.951 | 0.931 |
>
> Broad optimum at 0.3–0.7. Too small: lacks boundary information; too large: drifts beyond uncertainty region.
>
> Key Question 4:
>
> Yes. Results under varying $\lambda_3$:
>
> | $\lambda_3$ | 0.5 | 1.0 | 5.0 | 10.0 |
> |---|--:|--:|--:|--:|
> | Avg F1 | 0.957 | **0.960** | 0.924 | 0.885 |
> | Avg Precision | 0.952 | **0.958** | 0.887 | 0.814 |
> | Avg Recall | 0.962 | 0.962 | **0.967** | 0.969 |
>
> Excessive $\lambda_3$ causes sharp precision drops with marginal recall gains—boundary overfitting. Ablations confirm hard mining is auxiliary (~0.9% F1 drop when removed); $\lambda_3$ should remain moderate.

---

> > ### Author Rebuttal · Reviewer_H5uJ · 2026-04-03
> >
> > The authors have successfully resolved all my initial concerns by explicitly clarifying the structural distance metric, quantifying the inference speed trade-offs, providing detailed hyperparameter sensitivity analyses, and evaluating against recent baselines. As the theoretical and empirical justifications are now thoroughly validated with objective data, my previous reservations have been completely alleviated. Accordingly, I have upgraded my initial scores to reflect the confirmed soundness and practical viability of the proposed model.

---

### Official Review · Reviewer_am9r · 2026-03-12

**Soundness:** 3
**Presentation:** 2
**Significance:** 3
**Originality:** 3
**Overall Recommendation:** 4
**Confidence:** 4

**Summary:**

This paper proposes the MPFM (Cross Multi-Domain Prototype Flow Matching) framework for cross-domain log anomaly detection. Its core insight addresses a key limitation of existing methods: relying solely on geometric proximity to characterize normality fails to capture complex distribution boundaries in multi-domain scenarios. To address this, the authors introduce a dual-evidence paradigm that integrates three components: Shared-private prototypes to disentangle domain-invariant (shared) and domain-specific (private) patterns; Domain-conditional flow matching to model the generative dynamics of normal log data; Prototype drift-driven hard sample mining to strengthen the decision boundary for anomaly discrimination. Experiments on four log datasets (HDFS, BGL, Thunderbird, Spirit) demonstrate that MPFM outperforms both single-domain and transfer learning baselines.

**Compliance With Llm Reviewing Policy:**

Affirmed.

**Final Justification:**

My main concerns regarding negative transfer explanation, table formatting, LogDLR's cross-domain degradation mechanism, and LLM comparison have been well addressed in the rebuttal. I maintain my original score.

**Key Questions For Authors:**

A comparison of results in Table 1 and Table 2 shows that LogDLR (a baseline method) experiences a significant performance drop under cross-domain settings. What specific mechanisms in LogDLR’s design—particularly its global representation learning and domain adaptation modules—lead to this "interference" (i.e., performance degradation) in cross-domain scenarios?

Log anomaly detection is inherently a text-only task. Could the authors supplement their experiments with a performance comparison between MPFM and small-scale language models (small LLMs)? This would better contextualize MPFM’s competitiveness in text-centric log analysis and address whether LLMs (which excel at text pattern learning) offer comparable or superior performance.

**Limitations:**

No. The authors are encouraged to discuss the limitations of their work.

**Strengths And Weaknesses:**

Strengths

Clear Problem Articulation & Rational Technical Design: The paper clearly identifies two critical flaws of existing methods: (1) over-reliance on geometric distance for log anomaly detection, and (2) negative transfer caused by forced cross-domain alignment. It then proposes the DAMA (Dual-Evidence Matching Architecture) framework—combining structural evidence (from prototypes) and dynamic evidence (from flow matching)—to target these flaws. The problem statement is unambiguous, and the technical architecture is logically consistent.

Comprehensive Validation of Effectiveness: The paper supports the proposed method’s validity through both theoretical verification (to justify the dual-evidence paradigm) and extensive experimental evaluations (across multiple datasets and baselines), ensuring the reliability of its conclusions.

Weaknesses

Insufficient Clarity on Negative Transfer: While the paper repeatedly emphasizes that forced cross-domain alignment leads to negative transfer, it lacks direct and explicit explanations of two key points: (a) the specific causes of this negative transfer (e.g., whether it arises from conflicting domain-specific patterns or distorted invariant features), and (b) how the MPFM framework specifically mitigates this issue (e.g., which component—shared-private prototypes or flow matching—addresses negative transfer and why).

Misleading Formatting in Experimental Results: In Table 1, MPFM achieves significant performance improvements only on the Spirit dataset. However, incorrect bold formatting (applied to non-significant results for MPFM or baseline methods) misleads readers, obscuring the true extent of MPFM’s advantages and undermining the clarity of experimental findings.

---

> ### Author Rebuttal · Authors · 2026-03-30
>
> Weakness 1:
>
> We thank the reviewer for this comment. We will clarify the cause of negative transfer and the role of each component in the revision.
>
> (1) Cause of negative transfer. Normal patterns in multi-domain logs contain both cross-domain commonalities and domain-specific structures. Aggressive alignment that forces heterogeneous domains into an overly shared latent space suppresses domain-specific discriminative patterns and distorts domain manifold structures, causing pattern confusion and negative transfer.
>
> (2) How MPFM mitigates it. The shared–private prototype system is the primary component: shared prototypes capture transferable commonalities while private prototypes preserve domain-specific patterns, enabling knowledge sharing without overwriting domain information. The ablation study confirms this—removing private prototypes causes a maximum F1 drop of 3.6 pp on Spirit (average 2.6 pp across datasets). Domain-conditioned flow matching plays a complementary role by enforcing per-domain dynamical consistency constraints, allowing the model to reject samples that are geometrically close to prototypes yet dynamically inconsistent, thus reducing misalignment consequences. Hard example mining mainly strengthens robustness near decision boundaries rather than directly addressing negative transfer.
>
> Weakness 2:
>
> Thank you to the reviewer for pointing out this issue. As we were still revising and improving the manuscript up until the conference submission deadline, after the final round of edits we hastily submitted the paper without recompiling it, which resulted in the bold-formatting errors in Table 1 and the duplicated row entries in Table 2. We sincerely apologize for this oversight.
>
> In the revised manuscript, we will carefully review the entire paper and revise the corresponding discussions to ensure that every conclusion is strictly consistent with the corrected results. In particular, we will describe the extent of MPFM’s improvements more accurately—for example, in Table 1, MPFM’s significant advantage is mainly reflected on the Spirit dataset, rather than implying a uniform improvement across all datasets.
>
> Key Questions 1:
>
> Thank you for the reviewer’s question. The core idea of LogDLR is to learn a single domain-invariant latent space through a shared autoencoder and a domain discriminator with a gradient reversal layer. In multi-domain joint training, however, this may lead to two problems: (1) global representation learning compresses heterogeneous logs into a single shared space, which in multi-domain settings tends to capture globally common patterns while weakening normal structures that are valid only in specific domains, thereby blurring the anomaly boundary; and (2) domain-adversarial adaptation explicitly removes domain-discriminative features, which in highly heterogeneous multi-domain scenarios may cause over-alignment, suppressing domain-specific yet anomaly-relevant features as if they were nuisances, and thus leading to negative transfer.
>
> This is precisely why we introduce the shared-private prototype mechanism: it enables cross-domain knowledge sharing while explicitly preserving domain-specific structures, rather than forcing all domains into a fully aligned representation space.
>
> Key Questions 2:
>
> Thank you for the reviewer’s suggestion. Our current baselines already include the BERT-based LogBERT. To further address the reviewer’s concern regarding the performance of LLMs on this task, we additionally included comparisons with language-model-based baselines, including the encoder-style LogFiT, the prompt-based LogGPT, and the general small-scale large language models Baichuan and ChatGLM2, as shown in the table. It should be noted that all LLM baselines are trained and evaluated independently in a single-domain setting, whereas MPFM is jointly trained as a single unified model. Even under this stricter comparison setting, which is less favorable to MPFM, MPFM still significantly outperforms the LLM-based methods on most of comparable datasets. This demonstrates that the combination of flow matching and prototype mechanisms is better suited to cross-domain log anomaly detection than pure text modeling.
>
> | Method              | Type                             | HDFS  | BGL   | TB    | Spirit |
> | ------------------- | -------------------------------- | ----- | ----- | ----- | ------ |
> | LogFiT              | Encoder LM (single-domain)       | 0.950 | 0.899 | 0.939 | —      |
> | LogGPT (few-shot)   | Prompt-based LLM (single-domain) | —     | 0.625 | —     | 0.740  |
> | Baichuan (few-shot) | General LLM (single-domain)      | —     | 0.325 | —     | 0.527  |
> | ChatGLM2 (few-shot) | General LLM (single-domain)      | —     | 0.201 | —     | 0.137  |
> | MPFM (Ours)         | Multi-domain joint training      | 0.987 | 0.959 | 0.915 | 0.977  |

---

> > ### Author Rebuttal · Reviewer_am9r · 2026-04-03
> >
> > My concerns regarding negative transfer explanation, table formatting, LogDLR's cross-domain degradation mechanism, and LLM comparison have all been adequately addressed in the rebuttal. I will maintain my original score.

---

### Decision · Program_Chairs · 2026-04-30

**Decision:**

Accept (regular)

**Comment:**

This paper studies cross multi-domain log anomaly detection and proposes MPFM, which combines a shared–private prototype architecture with domain-conditioned flow matching and drift-driven hard example mining to address two key challenges in joint multi-domain learning: the insufficiency of geometric proximity alone and negative transfer under forced alignment. Reviewers agreed that the problem is practically important and that the method is conceptually coherent, with several noting that the dual structural/dynamical evidence design is well motivated. The main concerns focused on fairness of comparisons under joint training, reporting inconsistencies in the original tables, runtime overhead from ODE-based inference, sensitivity to hyperparameters, and the need for broader contextualization against LLM-based or recent cross-domain baselines. The rebuttal substantially addressed these issues by clarifying the training/inference design, correcting the interpretation of the comparison tables, adding efficiency and ODE-step trade-off results, reporting hyperparameter and seed sensitivity analyses, and including additional baseline comparisons. While I agree that some comparisons should be presented more carefully in the final version and that parts of the theory are better viewed as motivation than a complete formal analysis, I find the paper technically sound overall and the empirical case sufficiently strengthened after rebuttal. I therefore recommend accept.